# Rapid purification and metabolomic profiling of synaptic vesicles from mammalian brain

Lynne Chantranupong[1], Jessica L Saulnier[1], Wengang Wang[1], Drew R Jones[2†], Michael E Pacold[3†], Bernardo L Sabatini[1]*

[1]Department of Neurobiology, Howard Hughes Medical Institute, Harvard Medical School, Boston, United States; [2]New York University School of Medicine, Metabolomics Core Resource Laboratory at NYU Langone Health, New York, United States; [3]Department of Radiation Oncology, New York University Langone Medical Center, New York, United States

**Abstract** Neurons communicate by the activity-dependent release of small-molecule neurotransmitters packaged into synaptic vesicles (SVs). Although many molecules have been identified as neurotransmitters, technical limitations have precluded a full metabolomic analysis of SV content. Here, we present a workflow to rapidly isolate SVs and to interrogate their metabolic contents at high-resolution using mass spectrometry. We validated the enrichment of glutamate in SVs of primary cortical neurons using targeted polar metabolomics. Unbiased and extensive global profiling of SVs isolated from these neurons revealed that the only detectable polar metabolites they contain are the established neurotransmitters glutamate and GABA. In addition, we adapted the approach to enable quick capture of SVs directly from brain tissue and determined the neurotransmitter profiles of diverse brain regions in a cell-type-specific manner. The speed, robustness, and precision of this method to interrogate SV contents will facilitate novel insights into the chemical basis of neurotransmission.

*For correspondence:
bsabatini@hms.harvard.edu

†These authors contributed equally to this work

Competing interests: The authors declare that no competing interests exist.

## Introduction

Critical to the function of the brain are neurotransmitters, a diverse class of small molecules that act as chemical messengers between neurons. Neurotransmitters are stored in synaptic vesicles (SVs), membrane-bound organelles located within presynaptic axon terminals and whose activity-dependent release is essential for proper transmission of information within the brain (*Jahn and Südhof, 1994*). Upon electrical excitation of a neuron via action potentials, SVs rapidly fuse with the membrane to release their neurotransmitters, which are detected by transmembrane receptors on a postsynaptic neuron (*Südhof, 2013*; *Traynelis et al., 2014*). This binding event can trigger diverse consequences to the postsynaptic neuron, depending on the identity of the neurotransmitter and the receptors to which it binds. Whereas some neurotransmitters cause acute electrical activation or inhibition of a neuron by opening ion channels, others result in longer-term modulation of its signaling network by activating G-protein coupled receptors (*Nicoll et al., 1990*). Thus, the functional contribution of a neuron to a circuit is defined by the neurotransmitters it releases and the postsynaptic cells that it contacts.

Although neurotransmitters were discovered more than a century ago, our understanding of what molecules are used as neurotransmitters and how they change during life is likely incomplete. Neurons were classically believed to release only one fast-acting neurotransmitter, whose identity was fixed throughout the lifetime of the neuron (*Strata and Harvey, 1999*). However, neurons in many brain regions have recently been discovered to release multiple neurotransmitters (*Hnasko and*

*Edwards, 2012*; *Jonas et al., 1998*; *Root et al., 2014*; *Shabel et al., 2014*; *Tritsch et al., 2012*). Further increasing the complexity, these neurotransmitters may be packaged within the same or different SV pools or released from distinct axon terminals (*Hnasko and Edwards, 2012*; *Saunders et al., 2015*). Each possibility has unique effects on functionality and plasticity within circuits. In addition, neurons in the developing and mature brain have been found to lose, add, or switch the neurotransmitters they release in an activity-dependent manner (*Dulcis et al., 2013*; *Spitzer, 2012*). Finally, synapses exist in which the neurotransmitters released remain unknown. Although many neurons have been found to contain the machinery to synthesize and potentially release neurotransmitters, it remains to be established if this release occurs and whether it has functional consequences (*Mickelsen et al., 2017*; *Trapp and Cork, 2015*). Altogether, these discoveries greatly expand how neurotransmitters control brain circuitry and reveal the complexities that remain to be deduced.

Current techniques to infer the neurotransmitter identity of neurons rely on detecting the molecular machinery involved in synthesizing, packaging, or binding to neurotransmitters; however, these methods have several caveats which limit their applicability. Low mRNA and protein expression levels coupled with poor reagents for detection lead to false negatives and inaccurate conclusions (*Hnasko and Edwards, 2012*). Moreover, these approaches do not account for neurons that use unknown neurotransmitters or non-canonical pathways for neurotransmitter synthesis (*Kim et al., 2015*; *Tritsch et al., 2012*). In addition, the substrate specificities for many neurotransmitter receptors and vesicular transporters remain unclear (*Yelin and Schuldiner, 1995*). Alternatively, neurotransmitter identity is often inferred by pharmacological analysis of the receptors that mediate postsynaptic effects; for instance, a synaptic current is assumed to be induced by GABA release if it is blocked by an antagonist of ionotropic GABA receptors. However, many neurotransmitter receptors can be activated or allosterically modulated by diverse sets of small molecules that are found within cells, making this pharmacological approach difficult to interpret (*Macdonald and Olsen, 1994*; *Patneau and Mayer, 1990*).

Many of these concerns can be addressed by direct profiling of SV contents using mass spectrometry (MS), a powerful tool that identifies diverse metabolites in a systematic, sensitive and robust manner (*Patti et al., 2012*). Indeed, MS has greatly expanded our understanding of organellar biology by providing insight into their rich and dynamic metabolomes (*Abu-Remaileh et al., 2017*; *Chen et al., 2016*). To accurately profile the metabolic contents of SVs, quick and specific purification methods for SVs are required. However, current protocols are optimized for proteomic characterizations of SVs and require several hours to days to complete, during which the activity of transporters and biosynthetic enzymes may alter the contents of purified SVs (*Ahmed et al., 2013*; *Chen et al., 2016*). More importantly, these protocols are difficult to apply to specific neuronal populations in complex tissue.

To overcome these caveats, we developed a method to rapidly immunopurify SVs from both cultured mouse neurons and intact mouse brains within half an hour. In combination, we employed MS to directly and comprehensively interrogate the metabolic contents of SVs. With this workflow, we characterized the neurotransmitter profiles for diverse brain regions in a cell-type- specific manner. This method will serve as a foundation to discover and understand the diverse ways that neurons communicate with one another to control brain function.

## Results

### A method for rapid and specific capture of SVs from cultured neurons

In developing a method to dramatically reduce SV purification times while maintaining purity, we were inspired by immunoprecipitation (IP)-based workflows for organellar isolation which are highly specific and do not require time-consuming differential centrifugation techniques classically used to isolate organelles (*Abu-Remaileh et al., 2017*; *Chen et al., 2016*; *Ray et al., 2020*). We developed SV-tag, a construct in which a hemagglutinin (HA) tag is fused to the C-terminus of synaptophysin, an SV-specific, integral membrane protein and an ideal candidate to tag due to its ubiquitous presence and high abundance on SVs (*Figure 1A*; *Takamori et al., 2006*). In addition, we appended a tdTomato sequence followed by a self-cleaving T2A sequence to allow quick identification of infected cells via florescence. This strategy has several advantages compared to using an antibody

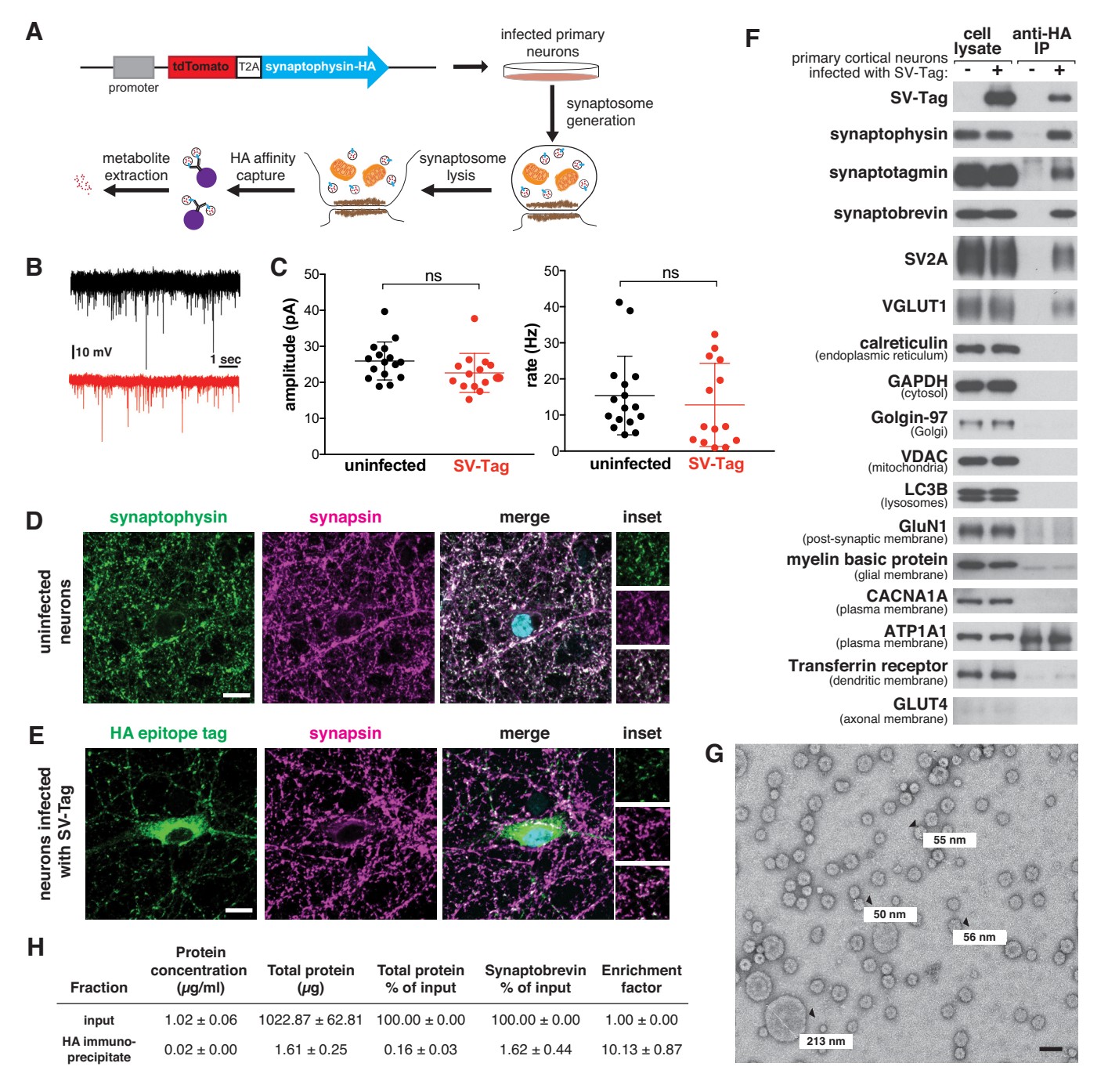

**Figure 1.** A method for rapid and specific isolation of synaptic vesicles (SVs) from mouse primary cortical cultures. (A) Construct design for tagging SVs and schematic of the workflow used to isolate SVs. (B) Representative traces of mEPSCs in uninfected neurons (black) and neurons infected with SV-tag (red). (C) Summary of the average amplitude (± standard deviation (std) and rate of mEPSCs in uninfected neurons and neurons infected with SV-tag ($V_{hold}$ = −70 mV, 1 μM TTX, 10 μM gabazine)). Non-significant p-value = n.s. (D) Immunostaining of uninfected primary neurons for endogenous synaptophysin (green) and synapsin (magenta). Cyan in the merged image represents DAPI-stained nuclei. Insets show selected fields that were magnified 1.6X. Scale bars: 10 μm. (E) Immunostaining of infected primary neurons expressing SV-tag (green) and synapsin (magenta) in. Insets show selected fields that were magnified 1.6X. Scale bars: 10 μm. (F) Immunoblot analysis of protein markers for SVs and indicated subcellular compartments and membranes in whole-cell lysates, purified SVs, and control immunoprecipitates. Lysates were prepared from neurons infected with lentivirus encoding SV-tag. 0.4% of the lysate and 5% of the immunoprecipitates were loaded into the indicated lane. (G) Electron microscope image of vesicles isolated with the workflow. Values denote diameter of indicated particles, specified by black arrows. Scale bar: 100 nm (H) Table summarizing the

*Figure 1 continued on next page*

Figure 1 continued

relative enrichment of synaptobrevin in the final immunoisolate from SV-tagged neurons, as assessed by quantitative immunoblotting. Values represent the mean ± std of three biological replicates. Source data is included (*Figure 1—source data 1*).

The online version of this article includes the following source data and figure supplement(s) for figure 1:

**Source data 1.** Quantification of total protein and synaptobrevin content in input and HA immunoisolates from the SV-tag workflow.
**Source data 2.** Analysis of particle size distribution in EM images of the final immunoisolate from SV-tagged neurons.
**Figure supplement 1.** Characterization of synaptic vesicles isolated from mouse primary cortical cultures.

against endogenous synaptophysin to isolate SVs. First, the SV-tag construct can be easily modified to express in genetically-defined subpopulations, which is crucial given the heterogeneity of neurons and brain tissue. Furthermore, the HA antibody is highly specific, sensitive, and well-characterized. Finally, the SV-tag strategy enables this method to be generalizable as the HA-tag is easily appended to other proteins and is compatible with many other applications, including metabolic and proteomic studies (*Chen et al., 2017*; *Huttlin et al., 2015*).

Because our workflow relies on immuno-affinity purification of subcellular compartments labeled with an ectopically expressed construct, it is necessary to ensure that this fusion protein does not perturb neurotransmitter release and is properly localized to SVs. In cultured mouse cortical neurons, the expression of SV-tag did not alter glutamate release from these cells, as determined by recording spontaneous miniature excitatory post-synaptic currents (mEPSCs) at a holding potential of $-70$ mV in the presence of TTX and gabazine (*Figure 1B–C*). To assess the localization of SV-tag, we compared its distribution to that of synapsin-1 and bassoon, both markers of presynaptic boutons (*De Camilli et al., 1983*; *Figure 1D and E*, *Figure 1—figure supplement 1A*). Although SV-tag colocalizes with synapsin and bassoon, a fraction of it is also detected in the soma in apparently synapsin-free areas (*Figure 1E*). In contrast, endogenous synaptophysin completely colocalizes with synapsin, with a small fraction of the signal in the perinuclear area (*Figure 1D*). We therefore pursued multiple optimization routes to improve the targeting of SV-tag. We moved the HA-tag to the N-terminus, lowered expression levels, tagged endogenous synaptophysin using CRISPR, tested other epitope tags (FLAG, GFP), and tagged other SV resident proteins (SV2A, VAMP2, synaptotagmin) (*Figure 1—figure supplement 1B*). Surprisingly, for all of these approaches the epitope tagged protein exhibited somatic localization comparable to that of SV-tag (*Figure 1—figure supplement 1B*). This suggests that for recombinant SV proteins, it is difficult to achieve the correct level of expression to ensure that they are trafficked from the soma to the boutons. Alternatively, it could indicate that a population of endogenous synaptophysin is somatically localized but not accessible by the synaptophysin antibody. This latter hypothesis is supported by the appearance of a somatic pool of synaptophysin when an HA-tag was introduced into the endogenous gene (*Figure 1—figure supplement 1B*, top row). Therefore, we decided to use our original synaptophysin-based SV-tag construct, due to the advantages of synaptophysin being a protein that is easy to express, abundant, and ubiquitously present on SVs (*Takamori et al., 2006*).

Based on a series of classical SV purification methods (*De Camilli et al., 1983*; *Craige et al., 2004*; *Huttner et al., 1983*; *Nagy et al., 1976*), we developed a workflow to immunoisolate SVs from cultured cortical neurons within 30 min, a substantial reduction in time compared to the multiple hours to days needed for classical methods (*Figure 1A*). First, we formed synaptosomes, which are isolated synaptic terminals (*Figure 1—figure supplement 1C*). Unlike direct cell lysis to release SVs, this process affords us a key advantage of separating and discarding the soma, along with its mislocalized SV-tag. We isolated synaptosomes within just seven minutes by optimizing the homogenization steps needed to generate synaptosomes and the speed and duration of spins that are necessary to separate unlysed cells from synaptosomes. ATP was added throughout the purification to maintain vATPase function, which establishes the proton gradient across the SV membrane that is necessary for the import of neurotransmitters and the maintenance of their levels within SVs (*Burger et al., 1989*). Following hypotonic lysis of the synaptosomes to release SVs, we immunoprecipitated SVs using HA antibodies conjugated to solid magnetic beads. The reduced porosity and magnetic properties of these beads enable cleaner, quicker capture compared to standard agarose beads, which can trap metabolite contaminants within their porous bead matrix (*Chen et al., 2016*). A series of high-salt washes post-immunoprecipitation disrupted non-specific protein and metabolite

interactions and further reduced contaminants. During our initial attempts to IP SVs, we used a triple HA-tag. Although this was sufficient to capture SVs as assessed by immunoblotting, the yield was low (*Figure 1—figure supplement 1D*). We reasoned that additional repeats of the HA epitope would increase capture efficiency by enhancing the probability that the tag will encounter an antibody during the IP period. Indeed, an extended tag of nine tandem HA sequences increased the yield of SVs despite expressing at a lower level than the triple HA-tag (*Figure 1—figure supplement 1D*).

To assess the quality and integrity of SVs isolated by this rapid procedure, we characterized the isolate remaining at the end of the purification for multiple features of SVs. Using immunoblotting, we confirmed the enrichment of SV protein markers such as synaptotagmin and SV2A and the concomitant depletion of markers for other subcellular organelles, compartments, and membranes (*Figure 1F*). One exception was the slight enrichment of dendritic endosomal membranes, as marked by the presence of transferrin receptor in our isolates. Improvements in the localization of SV-tag may help to alleviate this non-specific capture. We also used mass spectrometry to profile the proteome of isolated SVs in depth and found that the most significantly enriched proteins were SV resident proteins, including glutamate transporters, synaptobrevin, and vATPase subunits (*Supplementary file 1*; *Figure 1—figure supplement 1E*; *Grønborg et al., 2010*; *Takamori et al., 2006*). Furthermore, transmission electron microscopy revealed that the majority of particles isolated from our workflow are spheres of ~40–70 nm in diameter (*Eshkind and Leube, 1995*), as expected for SVs (*Figure 1G*, *Figure 1—figure supplement 1F*). In addition, we observed particles less than 40 nm, which may be membrane fragments resulting from high-salt washes performed at the end of the isolation to reduce contaminants. A minority of larger particles ($\geq$100 nm) are also present, which may be large dense-core vesicles (*Gondré-lewis et al., 2012*) or contaminating organelles and cellular debris. Finally, SVs isolated with SV-tag are comparable to those purified via lengthier, traditional differential centrifugation protocols, as assessed by immunoblot (*Figure 1—figure supplement 1H*) and electron microscopy (*Figure 1—figure supplement 1I*), albeit with a reduction in yield that comes as a tradeoff for the speed of isolation. To quantify yield of our workflow, we calculated how much of the SV material present in the input was captured in the immunoprecipitate. Based on three SV markers, our yields were ~1.5% (*Figure 1—figure supplement 1G*), which is consistent with previous rapid organellar isolations (*Chen et al., 2017*). Furthermore, we performed quantitative immunoblotting to determine that our preps were 10-fold enriched in SVs, as based on synaptobrevin (*Figure 1H*). This is lower than the 20X enrichment reported for previous isolation methods (*Ahmed et al., 2013*).

Although these analyses indicate we were able to enrich for SVs, they do not provide evidence that the SVs are intact, which is crucial for subsequent metabolite analysis. If the integrity of isolated SVs is not compromised, they should contain glutamate, the principal neurotransmitter of cultured cortical neurons (*Beaudoin et al., 2012*). Using a luminescence-based assay to detect glutamate, we observed an enrichment of glutamate in isolated SVs compared to the material obtained when the same protocol was applied to uninfected neurons (*Figure 1—figure supplement 1J*). Importantly, glutamate was depleted upon treatment of neurons with BafilomycinA (BafA), a vATPase inhibitor that dissipates the proton gradient of SVs that is essential for the import of glutamate into SVs (*Bowman et al., 1988*). In our isolation, it was not necessary to supplement ATP with magnesium (*Figure 1—figure supplement 1K*), likely due to the rapidity of the method and/or sufficient levels of magnesium released during cell lysis to maintain vATPase function. Taken together, multiple lines of evidence demonstrate that our SV-tag workflow enables rapid and specific high-affinity capture of intact SVs.

## Targeted and global metabolite profile of SVs from cultured neurons

To interrogate the metabolite contents of SVs in a precise and robust manner, we initially used targeted gas chromatography-mass spectrometry (GC/MS), a method with high sensitivity for many analytes, low cost and ease of operation, all of which are important considerations for optimization studies (*Beale et al., 2018*). We selected a panel of amino acids to profile, which included bona fide neurotransmitters (glutamate and glycine) (*Gundersen et al., 2005*), a putative neurotransmitter (aspartate) (*Fleck and Palmerv, 1993*), and non-neurotransmitter amino acids to assess the cleanliness of our preps. To identify metabolites that are enriched in SVs, we compared the signal of metabolites present in HA immunoprecipitates from SV-tag infected neurons vs. the signal from

immunoprecipitates of uninfected neurons, which served as a control for metabolites that non-specifically adhere to HA beads (*Chen et al., 2017*; *Supplementary file 3*). In primary cortical cultures, glutamate was the sole metabolite that was significantly enriched in SVs when compared to control (*Figure 2A*) and it was the only metabolite depleted by BafA treatment (*Figure 2B*), consistent with the excitatory and glutamatergic identity of these neurons. Importantly, non-neurotransmitter amino acids were not enriched, demonstrating that our SV preparations are of high purity as they lack contaminating metabolites. Of note, aspartate was not detected in these vesicles, suggesting that it does not function as a neurotransmitter in these cells. The same conclusion was reached via electrophysiological studies for hippocampal excitatory synapses (*Herring et al., 2015*).

To demonstrate the applicability of this method to profile other neuron types, we isolated SVs from inhibitory neuron cultures (*Figure 2—figure supplement 1A*), prepared from the medial ganglionic eminence (MGE) (*Franchi et al., 2018*). Unlike cortical cultures, these neurons are

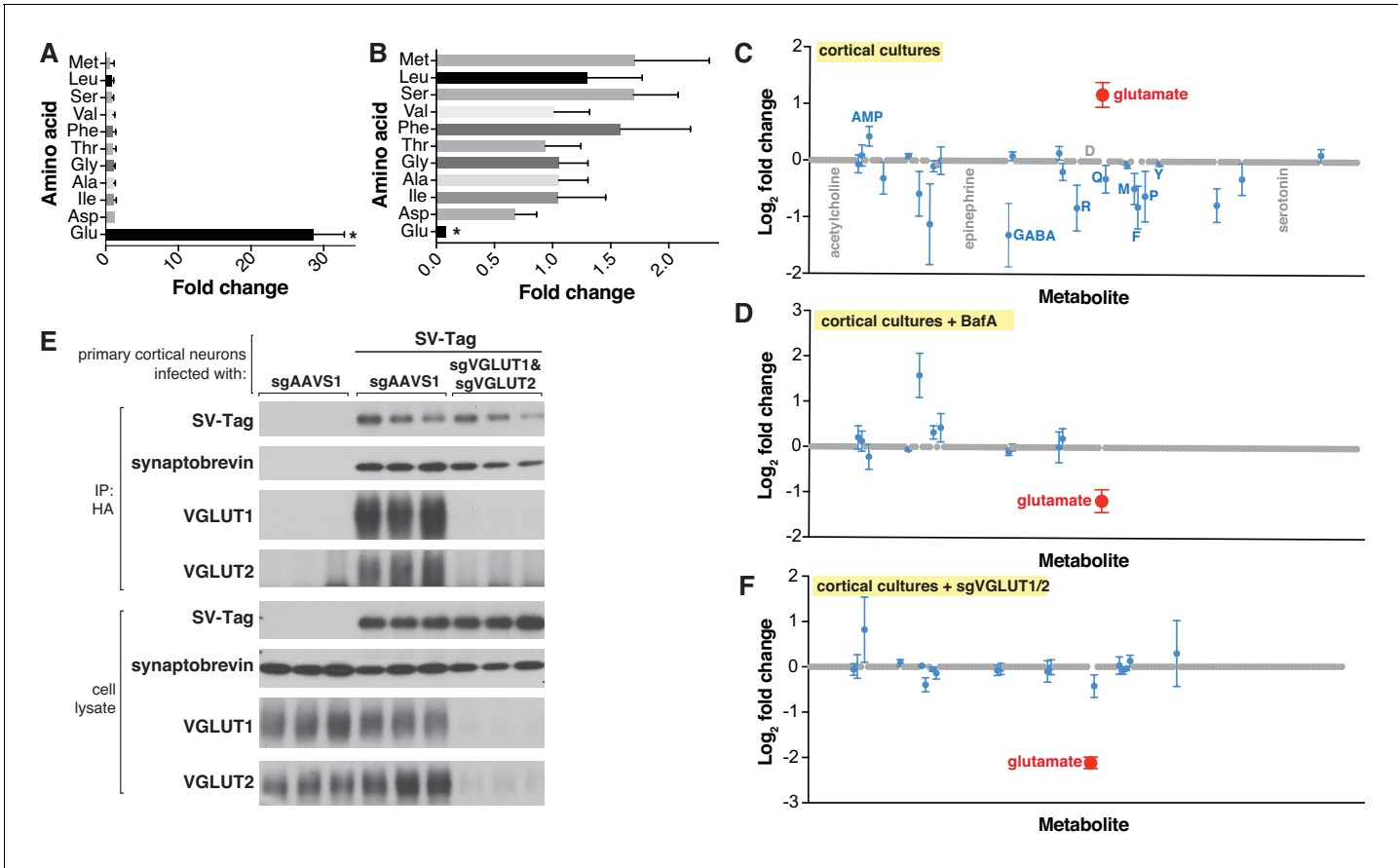

**Figure 2.** Targeted metabolite profile of purified synaptic vesicles (SVs) from cultured neurons. (A) Fold change (mean ± standard error of the mean (SEM), n = 3) of selected amino acids detected by GC/MS in purified SVs vs. control IPs. Asterisk denotes a statistically significant difference (p-value < 0.05) of the abundance of the indicated amino acid in SVs profiled from cells infected with SV-tag compared to uninfected cells. (B) Effect of pretreatment of neurons with BafilomycinA on the abundance of selected amino acids in purified SVs, as detected with GC/MS (mean ± SEM, n = 3) (C) Relative abundance via LC/MS of 153 polar metabolites present in purified SVs derived from SV-tag infected cells, compared to preps from uninfected cells (mean ± SEM, n = 3–4). Red indicates p-value < 0.05, blue indicates p- value > 0.05, and gray indicates that the metabolite was undetected in all samples. Single letter codes annotate selected amino acids. Metabolites are listed in alphabetical order, and their corresponding identities can be found in *Supplementary file 2*. (D) Effect of BafilomycinA on the presence of a panel of polar metabolites in purified SVs profiled with LC/MS (mean ± SEM, n = 3–4). Fold changes are color coded using the same specifications as in (C). (E) Immunoblot analysis of neurons expressing control guides (sgAAVS1) or guides targeting glutamate transporters (sgVGLUT1 and sgVGLUT2). Lysates were prepared from neurons infected with lentivirus encoding the indicated constructs. Fold changes are color coded using the same specifications as in (C). (F) LC/MS metabolite profile of SVs isolated from cells with glutamate transporter knockdown compared with control cells expressing the control guide.

The online version of this article includes the following figure supplement(s) for figure 2:

**Figure supplement 1.** Comparison of metabolite profiles of isolated synaptic vesicles from cultured excitatory and inbitory neuronns.

GABAergic, as evidenced by their expression of VGAT, a GABA transporter (*Wojcik et al., 2006*), and their lack of VGLUT1 protein, a glutamate transporter (*Pines et al., 1992*; *Figure 2—figure supplement 1B*). Reflecting the differences in the neurotransmitter identities of cortical and MGE-derived neurons, GC/MS analysis revealed that SVs isolated from MGE cells are enriched for GABA but not glutamate, whereas the converse is observed for cortical cultures (*Figure 2—figure supplement 1C*). Highlighting the specificity of our method, MGE SVs were not significantly enriched for any other amino acids profiled. Thus, by combining the SV-tag isolation workflow with GC/MS, we can successfully obtain neurotransmitter profiles of diverse neuron subtypes.

Although GC/MS provides a strong starting point for analysis, it is limited in its capacity to detect a wide range of polar metabolites, due to the need for polar metabolite volatilization and separation on a GC column (*Lei et al., 2011*; *Sobolevsky et al., 2003*; *Stenerson, 2011*). Therefore, we transitioned to metabolomic analysis by liquid chromatography – high-resolution mass spectrometry (LC/MS), which can detect a much broader range of metabolites with nanomolar sensitivity (*Lei et al., 2011*). We quantified the relative abundance of 153 polar compounds, which included neurotransmitters such as acetylcholine, serotonin, and epinephrine, as well as key molecules in metabolic pathways and subcellular compartments (*Supplementary file 2*). Corroborating our earlier findings with GC/MS, glutamate was the sole detected metabolite that is significantly enriched within SV-tag isolated SVs (*Figure 2C*) and depleted upon BafA treatment (*Figure 2D*; *Supplementary file 5*). All other metabolites were either undetected or detected at statistically insignificant levels. Reflecting the specificity of our workflow for isolating SVs, markers for other subcellular compartments were not enriched, including cystine, which is characteristic of lysosomes (*Pisoni and Thoene, 1991*), and aspartate and phosphocholine, which are the most enriched metabolites in mitochondria (*Chen et al., 2016*).

Because vATPase resides in other organelles (*Futai et al., 2000*), the observation that its inhibition depletes glutamate is not sufficient to provide certainty that the glutamate detected is within SVs. To address this concern, we took advantage of the fact that glutamate is imported into SVs via glutamate transporters VGLUT1 and VGLUT2 (*Takamori et al., 2001*), which, unlike vATPase, are SV-specific proteins (*Fremeau et al., 2001*). Reassuringly, CRISPR-mediated depletion of these transporters from cortical neurons by targeting the genes *Slc17a6* (VGLUT2) and *Slc17a7* (VGLUT1) significantly reduced glutamate levels from isolated SVs (*Figure 2E and F*). Taken together, these results demonstrate that our purification workflow isolates SVs of high purity and integrity which are compatible for robust and sensitive profiling with multiple MS methods.

Previous work has proposed that other polar molecules may function as neurotransmitters, including taurine, gamma-hydroxybutyrate, ß-alanine, and agmatine (*Cash, 1994*; *Davison and Kaczmarek, 1971*; *Kilb and Fukuda, 2017*; *Reis and Regunathan, 2000*; *Tiedje et al., 2010*). To gain a more comprehensive and unbiased view of the SV metabolome, we performed a global metabolomics screen for polar metabolites (*Figure 3A–B*). The LC/MS runs were optimized for detection of a broad range of polar retentivities using a HILIC stationary phase and fast polarity- switching MS method to capture analytes which ionize exclusively in positive or negative mode. Overall, we detected 2724 representative features which were defined by an observed m/z (25 ppm) at a specific retention time (±0.25 min) with a minimum intensity and signal to noise threshold (*Supplementary file 4*).

To identify SV-specific metabolites, we filtered for peaks that were significantly enriched by at least two-fold in SVs and concomitantly depleted by BafA by at least two-fold. To assign the peak identity of this small subset of features, we used a variety of manual approaches including spectral library search, accurate mass formula search, and isotope fine structure. To our surprise, only three metabolites satisfied these criteria – glutamate, GABA, and potassium (*Figure 3C*, *Supplementary file 4*). In these samples, GABA likely originates from a minority population of interneurons in cultured cortical neurons, consistent with the low expression level of VGAT, the GABA transporter, observed in these cultures (Figure S2B). The variability of the number of GABAergic interneurons across different cortical culture preps likely contributes to the variation in the GABA levels detected across MS runs. In addition to GABA, we identified a peak that is associated with potassium, which arises from the putative pairing of this ion with carbonate ions in the LC buffers used in the IP/MS workflow. However, potassium is a ubiquitous component of multiple reagents and sample-to-sample fluctuations in their levels can contribute to altered potassium content. Follow-up studies using atomic absorption spectroscopy or inductively coupled plasma mass

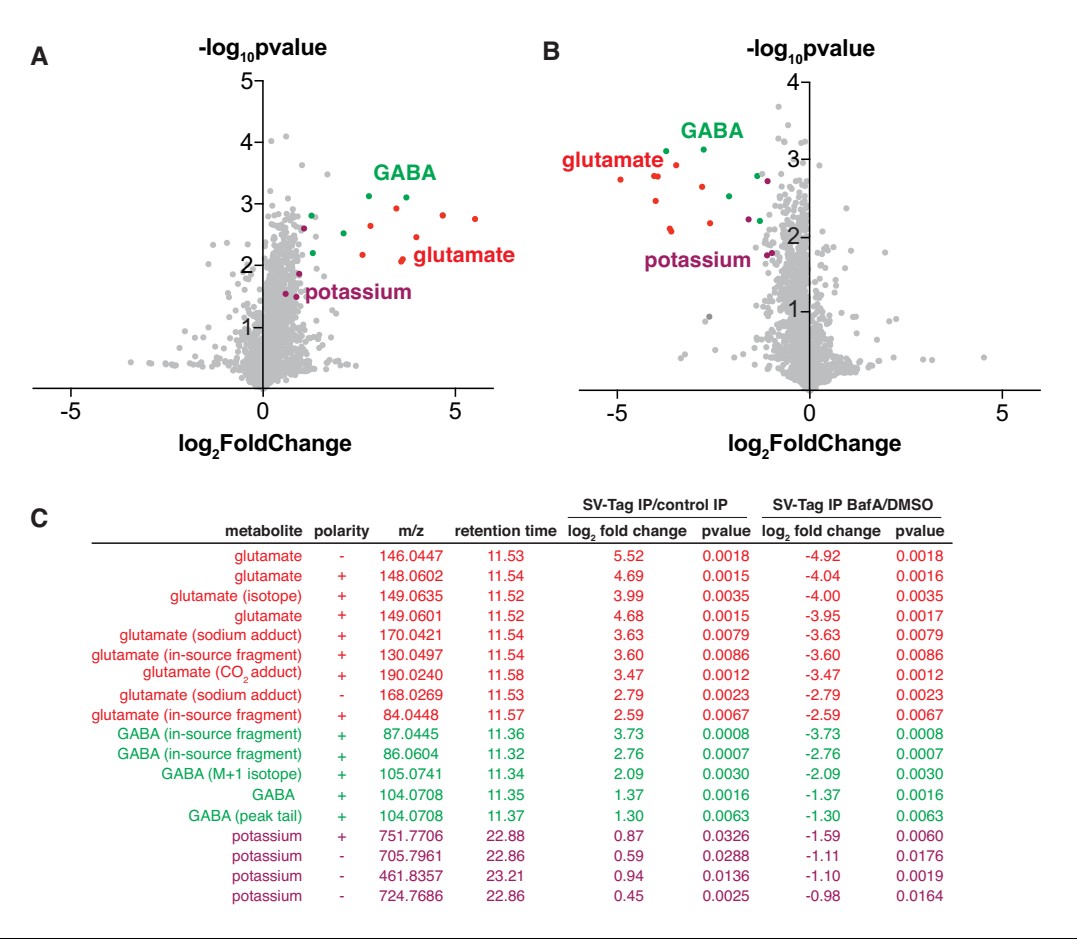

**Figure 3.** Unbiased polar metabolomics profile of purified synaptic vesicles (SVs) from cultured cortical neurons. (**A**) Global polar metabolomics analysis via LC/MS of purified SVs, compared to an IP from uninfected cells. Green indicates glutamate and its associated derivatives generated during the LC/MS run. Red indicates GABA and its derivatives. Purple indicates potassium. Each dot represents the average of three replicate samples. (**B**) Global polar metabolomics analysis on purified SVs from Bafilomycin-treated vs. DMSO treated neurons. Legend is same as in A. (**C**) Summary of metabolites from global analysis which are significantly enriched in SV-tagged SVs and significantly depleted by BafilomycinA treatment.

spectrometry (ICP-MS) will be necessary to establish how much potassium these SVs quantitatively contain, whether these weakly enriched potassium peaks are indeed internalized within SVs, and if they contribute to SV function.

## Adaptation of the method for SV isolation and profiling from brain tissue

Given their relative homogeneity of neuron types and ease of preparation, cultured cortical neurons provide an ideal setting to optimize and test purification protocols. However, in the brain, molecularly and functionally distinct neurons are intermingled in an intricate and heterogeneous environment, and they rely on the uptake of extracellular metabolites found in this environment for neurotransmitter synthesis (*Elsworth and Roth, 1997*; *Mathews and Diamond, 2003*; *Schousboe et al., 2013*). To gain a more complete understanding of neurotransmission and potentially identify unknown endogenous neurotransmitters, it is necessary to profile SVs isolated from their native environment. We therefore adapted the method for use in brain tissue (*Figure 4A*). First, we expressed SV-tag in the brain by transducing desired brain regions of mice via stereotaxic injections of adeno-associated viruses (AAV) encoding Cre-independent SV-tag. To ensure neuron-specific expression, the expression of SV-tag was driven by the synapsin promoter (*Kügler et al., 2003*). SV-tag readily expresses in diverse areas, as evidenced by the abundance of tdTomato positive neurons in targeted regions (*Figure 4B*). To ensure that SV-tag does not impair neurotransmission in

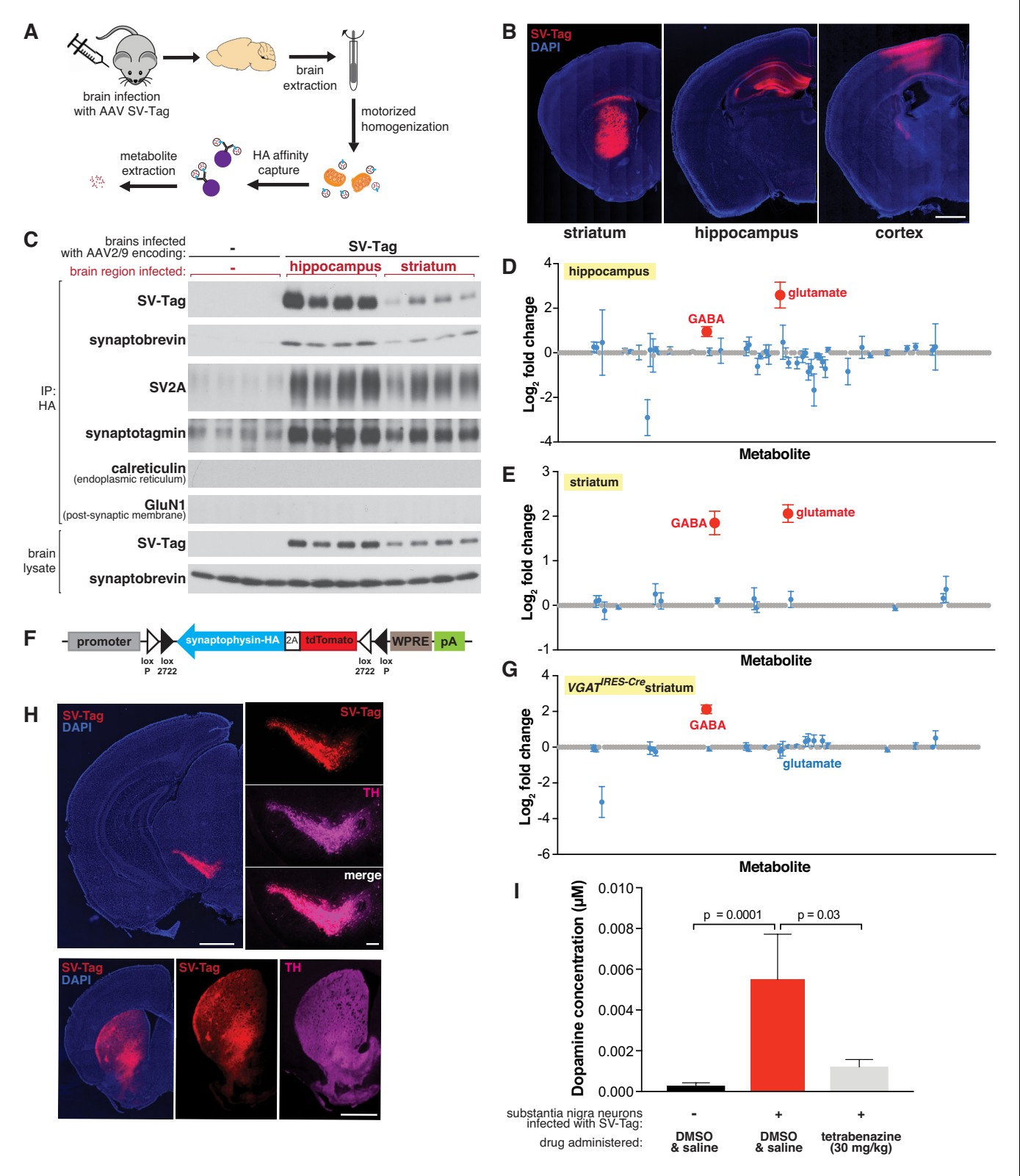

**Figure 4.** Adaptation of the workflow for rapid and specific isolation and metabolite profiling of synaptic vesicles (SVs) directly from mouse brain tissue. (A) Schematic of the workflow used to isolate SVs from mouse brain tissue. (B) Immunofluorescence images of coronal sections from wild-type mouse brains transduced with SV-tag in the indicated brain regions. Neurons are labeled with DAPI nuclear stain (blue) and SV-tag (red). Scale bar: 1 mm. (C) Immunoblot analysis of indicated protein markers present in brain lysates, control immunoprecipitates from uninfected brains, and HA

*Figure 4 continued on next page*

*Figure 4 continued*

immunoprecipitates from hippocampi and striatum that were infected with SV-tag. (D) LC/MS profile of SVs isolated from wild-type mice brains infected with SV-tag in hippocampus compared to a control IP from uninfected brains. (mean ± SEM, n = 4). Color code and legend is the same as in *Figure 2C*. (E) LC/MS profile of SVs isolated from wild-type mice brains infected with SV-tag in striatum compared to uninfected brains. (mean ± SEM, n = 4) (F) Construct design for expression of SV-tag in neurons in a Cre-dependent manner. (G) LC/MS profile of SVs isolated from *Slc32a1*[IRES-Cre/wt] (*VGAT*[IRES-Cre]) mice brains infected with SV-tag in striatum compared to uninfected brains. (mean ± SEM, n = 4) (H) Coronal sections from a *Slc6a3*[IRES-Cre/wt] mouse transduced with Cre-dependent SV-tag in dopaminergic neurons of the midbrain. Dopamine neurons are immunolabelled for tyrosine hydroxylase (TH, magenta), DAPI nuclear stain (blue) and SV-tag (red). (I) Targeted LC/MS profiling of dopamine in SVs isolated from *Slc6a3*[IRES-Cre/wt] mice transduced with Cre-dependent SV-tag in dopaminergic neurons of the midbrain. Indicated mice were subjected to saline injection or tetrabenazine injection intraperitoneally 2 hr prior to harvesting of SVs.

The online version of this article includes the following source data and figure supplement(s) for figure 4:

**Source data 1.** Absolute dopamine concentrations measured for *Figure 4I*.
**Figure supplement 1.** Characterization of synaptic vesicles isolated directly from mouse brain tissue.

vivo, we examined synaptic transmission at well-characterized Schaffer collateral synapses between hippocampal CA3 and CA1 pyramidal cells (*Figure 4—figure supplement 1A*; *Jackman et al., 2014*). Paired-pulse facilitation, which indicates changes in probability of release from presynaptic terminals, was assessed using pairs of closely spaced electrical stimuli (*Figure 4—figure supplement 1B*). Paired-pulse ratios, measured at an inter-stimulus interval of 50 ms, were unaffected by SV-tag expression in presynaptic neurons, which indicated that SV-tag did not significantly alter neurotransmitter release probability at this synapse (Figure S4C).

Our initial attempts to isolate SVs from brain tissue using the same strategy developed for cultured neurons resulted in preps with poor yield and high background. This is not surprising given the complex structure and composition of brain tissue compared to cultured neurons, which grow as a homogeneous monolayer. To resolve these issues, we compared several lysis protocols for their efficiency in releasing SVs from neurons. Compared to forming and lysing synaptosomes, homogenizing the whole brain to immediately free SVs greatly improved yields (*Figure 4—figure supplement 1D*). Importantly, a motorized homogenizer was necessary for effective and rapid lysis. The addition of a final five-minute high-salt incubation after the IP further reduced background contamination. In combination, these changes enabled SV isolation in under 30 min from diverse brain regions, including hippocampus, cortex, and striatum (*Figure 4C*, *Figure 4—figure supplement 1E*). Moreover, SVs had minimal contamination of other subcellular organelles as demonstrated by immunoblot and electron microscope analyses (*Figure 4C*, *Figure 4—figure supplement 1E, F*).

Utilizing the LC/MS workflow, we determined if our method is capable of distinguishing the content of SVs isolated from brain regions with different neurotransmitter profiles. We profiled SVs from the hippocampus, which is composed of roughly 90% glutamatergic pyramidal neurons and 10% GABAergic interneurons (*Olbrich and Braak, 1985*), and the striatum, which is comprised primarily of GABAergic cells (*Yoshida and Precht, 1971*). The metabolite profiles of hippocampal SVs are expected, with glutamate and GABA detected and glutamate being the more abundant neurotransmitter (*Figure 4D*). Similar to cortical culture SVs, no other metabolites other than these two known neurotransmitters were significantly present.

In contrast, striatal SVs were contaminated by a significant amount of glutamate. Because SV-tag was expressed in a Cre-independent manner in wild-type mice, this could result from isolation of SVs from glutamatergic cortical neurons that lie above striatum and became infected with SV-tag due to pipette withdrawal and viral leak or via retrograde transduction due to axonal-uptake of AAV. Alternatively, it could be due to contamination by cortical and thalamic glutamatergic terminals that heavily innervate the striatum. To distinguish between these possibilities, we generated a Cre-dependent construct of SV-tag to enable cell-type-specific expression of this IP handle (*Figure 4F*; *Atasoy et al., 2008*). Striatal SVs from *Slc32a1*[IRES-Cre] mice, where SV-tag expression is restricted to GABAergic neurons (*Vong et al., 2011*) and SVs from *Adora2a*[Cre] mice, which restrict expression further to a subpopulation of GABAergic striatal neurons (*Durieux et al., 2009*; *Figure 4—figure supplement 1G*), reveal the enrichment of GABA and the concomitant depletion of glutamate (*Figure 4G* and *Figure 4—figure supplement 1H*). Altogether, these results indicate that cell-type specific expression of SV-tag permits analysis of SV content on material isolated directly from complex brain tissue.

Thus far we focused on the detection of neurotransmitters whose polar and stable nature is compatible for sensitive detection by LC/MS. However, some neurotransmitters are less stable, such as the rapidly oxidizable dopamine, a crucial neurotransmitter whose loss underlies debilitating motor ailments such as Parkinson's disease (*Damier et al., 1999*). To test if, despite its labile nature, our rapid SV isolation workflow enables detection of dopamine, we expressed SV-tag in midbrain substantia nigra neurons of *Slc6a3^IRES-Cre* mice where Cre recombinase is restricted to cells expressing DAT, the plasma membrane dopamine transporter. As has been previously demonstrated in the injection target, Cre-expressing cells in this mouse line match well with the population of neurons that express VMAT2, the vesicular dopamine transporter (*Edwards, 1992*; *Lammel et al., 2015*). We validated specificity of expression by immunohistochemical analysis, which indicated selective expression of SV-tag in dopaminergic cells as seen by colocalization with tyrosine hydroxylase, a dopamine synthetic enzyme (*Daubner et al., 2011*) both in the soma and in axonal projections to striatum (*Figure 4H*). In a cohort of mice expressing SV-tag, we administered tetrabenazine, a reversible inhibitor of VMAT2 (*Erickson et al., 1996*; *Scherman et al., 1983*), at a concentration validated to deplete dopamine from SVs in acute striatal slices (*Figure 4—figure supplement 1I–J*). Using a modified LC/MS method for the detection of dopamine, we find that substantia nigra SVs are enriched for dopamine and these stores are depleted by VMAT2 inhibition (*Figure 4I*). Taken together, we developed a workflow to enable rapid and specific purification of SVs directly from the brain, and to obtain the neurotransmitter profiles of a wide range of neurons in a cell-type-specific manner.

## Discussion

Neurons use a broad array of small molecules for fast chemical neurotransmission. The ability for different neuron classes to release different transmitters, along with the diversity of neurotransmitter-specific postsynaptic receptors, enables complex and intermingled excitatory, inhibitory, and neuromodulatory trans-synaptic signaling in dense brain tissue. Furthermore, individual neurons are able to release multiple transmitters, sometimes releasing different neurotransmitters to different postsynaptic targets, as well as switch neurotransmitter identity in a developmental and activity dependent manner. This richness of neurotransmitter usage in the brain highlights the need for new methods to deduce neurotransmitter identity in an accurate and cell-type-specific manner. To address these needs, we developed a workflow that combines high-affinity capture of epitope tagged SVs with MS-based metabolomics to comprehensively profile SV contents. We adapted our method to isolate SVs with speed and ease from both primary cultured neurons and genetically-defined neurons in brain tissue, which will allow complementary in vitro and in vivo studies in the future. Our study confirms the utility of combining IP- and MS-based methods to characterize organelles such as SVs, and underscores the importance of performing global analyses to draw unbiased conclusions of the metabolic contents of organelles.

Our analyses of polar metabolites in SVs profiled in this study reveal that these organelles have a minimal metabolic content and contain only previously recognized *bona fide* neurotransmitters, which is consistent with their primary role in neurotransmitter storage and release. We find that glutamatergic vesicles, isolated from either cultured neurons or directly from the brain, contain only glutamate and lack detectable amounts of other proposed neurotransmitters (such as aspartate). Indeed, untargeted analysis of polar metabolites failed to detect any additional molecule that was enriched within these vesicles. It is possible that non-polar molecules and ions could also be imported into SVs. Indeed, zinc has been proposed to function as a neurotransmitter at excitatory synapses (*Vergnano et al., 2014*). In future studies, it will be beneficial to modify our workflow to be compatible with MS methods for atomic ion and non-polar metabolite detection. In addition, further optimization of the immunocapture procedure to increase SV yields is necessary to utilize MS methods for atomic ion detection, which require high amounts of starting material. Finally, further enhancements in the sensitivity of MS detection of metabolites coupled with more extensive profiling of other neuronal cell types in vivo may uncover putative neurotransmitters.

Although our method is rapid and specific, improvements to increase its sensitivity for neurotransmitter detection, particularly from SVs isolated from the brain, would be beneficial, even for the detection of polar metabolites. We were able to identify dopamine in SVs from substantia nigra neurons, but not GABA and glutamate (*Tritsch et al., 2016*; *Zhang et al., 2015*), which are known to be

released by these cells. In addition, we observed a robust GABA signal in SVs from striatal cells but did not detect acetylcholine, which is released by a small population of interneurons that reside in this region (*Cox and Witten, 2019*). As cholinergic neurons represent about 2% of striatal neurons (*Zhou et al., 2002*), this suggests a lower limit for the ability of our current protocol to detect minor neurotransmitters. The yield of SVs captured by our method and thus the detection limit would likely be increased by the use of a mouse line in which synaptophysin is endogenously tagged with HA in a conditional manner. This would permit more uniform and complete expression of SV-tag in desired cells without ectopic overexpression, and thus increase SV yields and the consistency between preps. Furthermore, it would eliminate the need for tedious injections and facilitate greater turnaround time of experiments.

With a method to identify molecules that are enriched in SVs, we can pursue many intriguing questions and avenues. To determine whether specific molecules function as neurotransmitters, candidates can be easily profiled for their presence inside SVs using our workflow. For instance, in the context of cultured neurons and the brain regions profiled in this study, neither aspartate nor taurine - molecules long debated to be neurotransmitters - were enriched within SVs, thus bringing into doubt their potential synaptic functions. Analysis in the hippocampus also failed to find evidence in support of aspartate being a neurotransmitter (*Herring et al., 2015*). Another important avenue of research is elucidating the function of transporter-like SV proteins such as SV2A and defining the substrate specificities of promiscuous vesicular transporters, including VMAT2 (*Lynch et al., 2004*; *Yelin and Schuldiner, 1995*). Loss of function experiments combined with global MS profiling will provide a powerful strategy to interrogate how these proteins affect SV function. Finally, this method can be easily adapted to other applications to resolve long-standing questions about SVs. For instance, the release properties of SVs have long been known to be heterogeneous, and only a fraction of SVs known as the rapidly releasable pool fuse upon electrical excitation of a neuron (*Rizzoli and Betz, 2005*). Proteomic profiling of these different SV populations following immunoisolation could reveal the molecular basis for these differences, which has long eluded researchers. Finally, this method can facilitate the identification of neurons which corelease multiple transmitters and follow-up studies with higher resolution methods such as EM or array tomography can address the basis of this corelease and if it occurs from the same or different vesicle. In combination, these possibilities highlight the potential of SV immunoisolation and metabolomics to address a wide range of biological questions.

In conclusion, the robustness and ease of our SV isolation and profiling methodology will serve as a platform upon which we can gain a deeper understanding of the diverse ways that these organelles control neurotransmission.

# Materials and methods

**Key resources table**

| Reagent type (species) or resource | Designation | Source or reference | Identifiers | Additional information |
|---|---|---|---|---|
| Strain, strain background (*M. musculus*) | Wild-type | Jackson Labs | C57BL6/J, RRID:MGI:5650797 | Stock #00644 |
| Strain, strain background (*M. musculus*) | Albino Swiss mice | Charles River | CD-1 IGS, RRID:MGI:5653285 | Stock #022 |
| Strain, strain background (*M. musculus*) | Vgat[ires-cre] | Jackson Labs | *Slc32a1*[IRES-Cre], RRID:IMSR_NM-KI-200081 | Stock #016962 |
| Strain, strain background (*M. musculus*) | Dat[ires-cre] | Jackson Labs | *Slc6a3*[IRES-Cre], RRID:IMSR_NM-KI-200092 | Stock #006660 |
| Strain, strain background (*M. musculus*) | *Adora2a*[Cre] | GENSAT | RRID:MMRRC_034744-UCD | founder line KG139 |
| Strain, strain background (AAV) | AAV2/9 hSynapsin SV-tag WPRE | BCH Viral Core | NA | Titer: $1.6 \times 10^{14}$ gc/ml |
| Strain, strain background (AAV) | AAV2/9 Cre floxed SV-tag WPRE | BCH Viral Core | NA | Titer: $1.7 \times 10^{14}$ gc/ml |

*Continued on next page*

*Continued*

| Reagent type (species) or resource | Designation | Source or reference | Identifiers | Additional information |
|---|---|---|---|---|
| Antibody | Guinea pig polyclonal synaptophysin | Synaptic Systems | 101004, RRID:AB_1210382 | (1:100) dilution |
| Antibody | Rabbit polyclonal synaptophysin | Synaptic Systems | 101002, RRID:AB_887905 | (1:1000) dilution |
| Antibody | Rabbit monoclonal synapsin | Cell Signaling Technology | 5297S, RRID:AB_261578 | (1:500) dilution |
| Antibody | Rabbit monoclonal HA-Tag | Cell Signaling Technology | 3724S, RRID:AB_1549585 | (1:1000) dilution |
| Antibody | Mouse monoclonal synaptotagmin | Synaptic Systems | 105011, RRID:AB_887832 | (1:1500) dilution |
| Antibody | Mouse monoclonal synaptobrevin | Synaptic Systems | 104211. RRID:AB_887811 | (1:5000) dilution |
| Antibody | Rabbit monoclonal SV2A | Synaptic Systems | 119003, RRID:AB_2725760 | (1:2000) dilution |
| Antibody | Rabbit polyclonal VGLUT1 | Synaptic Systems | 135303, RRID:AB_887875 | (1:1500) dilution |
| Antibody | Rabbit monoclonal VGLUT2 | Synaptic Systems | 135421, RRID:AB_2619823 | (1:1000) dilution |
| Antibody | Rabbit monoclonal calreticulin | Cell Signaling Technology | 12238S, RRID:AB_2688013 | (1:250) dilution |
| Antibody | Rabbit monoclonal GAPDH | Cell Signaling Technology | 2118S, RRID:AB_1031003 | (1:1000) dilution |
| Antibody | Rabbit monoclonal Golgin-97 | Cell Signaling Technology | 13192S, RRID:AB_2798144 | (1:200) dilution |
| Antibody | Rabbit monoclonal VDAC | Cell Signaling Technology | 4661S, RRID:AB_10557420 | (1:500) dilution |
| Antibody | Rabbit monoclonal LC3B | Cell Signaling Technology | 2775S, RRID:AB_915950 | (1:200) dilution |
| Antibody | Mouse monoclonal NMDA receptor | Synaptic Systems | 114011, RRID:AB_887750 | (1:1000) dilution |
| Antibody | Rabbit polyclonal Myelin basic protein | Synaptic Systems | 295003, RRID:AB_2620036 | (1:300) dilution |
| Antibody | Rabbit polyclonal $Ca^{2+}$ channel P/Q-type alpha1A unit | Synaptic Systems | 152103, RRID:AB_887699 | (1:300) dilution |
| Antibody | Mouse monoclonal apha1 $Na^+$ $K^+$ ATPase | Abcam | Ab7671, RRID:AB_306023 | (1:300) dilution |
| Antibody | Rabbit polyclonal Transferrin receptor | Abcam | Ab84036, RRID:AB_10673794 | (1:300) dilution |
| Antibody | Mouse monoclonal GLUT4 | Cell Signaling Technology | 2213S, RRID:AB_823508 | (1:200) dilution |
| Antibody | Mouse monoclonal tyrosine hydroxylase | ImmunoStar | 22941, RRID:AB_572268 | (1:1000) dilution |
| Antibody | Mouse monoclonal Bassoon | Enzo Life Sciences | SAP7F407, RRID:AB_1641480 | (1:1000) dilution |
| Antibody | Rabbit monoclonal VGAT | Synaptic Systems | 131011, RRID:AB_887872 | (1:1000) dilution |
| Peptide, recombinant protein | Recombinant mouse synatopbrevin protein (His tagged) | Abcam | Ab222979 | |
| Chemical compound, drug | BafilomycinA | Millipore Sigma | 196000 | |

*Continued on next page*

*Continued*

| Reagent type (species) or resource | Designation | Source or reference | Identifiers | Additional information |
|---|---|---|---|---|
| Chemical compound, drug | Tetrabenazine | Sigma | T2952-10MG | |
| Recombinant DNA reagent | pUbc SV-tag (lentiviral vector) | This study | | *Figure 1A* |
| Recombinant DNA reagent | pAAV hSynapsin SV-tag (AAV vector) | This study | | *Figure 4F* |
| Recombinant DNA reagent | pAAV hSynapsin flexed SV-tag (AAV vector) | This study | | *Figure 4F* |
| Recombinant DNA reagent | pLenti CRISPR sgVGLUT1 (lentiviral vector) | This study | | Methods section, 'Lentiviral constructs and production' |
| Recombinant DNA reagent | pLenti CRISPR sgVGLUT2 (lentiviral vector) | This study | | Methods section, 'Lentiviral constructs and production' |
| Commercial assay or kit | Glutamate- Glo assay | Promega | J7021 | |
| Other | Anti-HA magnetic beads | Thermo Fisher Scientific | 88837, RRID:AB_2861399 | |

## Preparation of neuronal cultures and drug treatments

Primary dissociated cortical cultures were prepared from cortices of E16-E18 embryos of CD-1 mice (Charles River stock # 022) as described (*Sciarretta and Minichiello, 2010*). Cultured inhibitory neurons were prepared from the MGE of E13-E14 mice. The following modifications were made to enhance culture health: tissue pieces were digested with papain (Worthington) instead of trypsin-EDTA, DNAse (Sigma) was added to break down released genomic DNA and aid in more efficient trituration, and 1 ml pipette tips (PipetOne) were used to titurate the tissue instead of fire polished pipettes for more consistency.

For immunohistochemistry experiments, ~1 million cells were plated onto 24-well plates pre-coated with Poly-D-Lysine (PDL) (Sigma) and laminin (Invitrogen). For biochemical isolations, 4–5 million cells were plated onto 10 cm plates pre-coated with Poly-D-Lysine (PDL) and laminin. Cells were maintained at 37°C and 5% $CO_2$. A third of the media was replaced every 3 days with fresh Neurobasal media, and 1 μM Arabinoside C (Sigma) was added on the fifth day in vitro (DIV5) to prevent overgrowth of astrocytes and microglia.

In experiments where SVs were isolated from BafilomycinA (BafA) treated neurons, prior to the isolation cells were treated for 2 hr with 500 nM BafA (EMD Millipore) or the corresponding DMSO volume.

## Lentiviral constructs and production

The following sense (S) and antisense (AS) oligonucleotides encoding the guide RNAs were cloned into a pLentiCRISPR vector (Addgene 52961):

    sgAAVS1 (S): caccgTCCCCTCCACCCCACAGTG
    sgAAVS1 (AS): aaacCACTGTGGGGTGGAGGGGAc
    sgVGLUT1 (S): caccgGGAGGAGTTTCGGAAGCTGG
    sgVGLUT1 (AS): aaacCCAGCTTCCGAAACTCCTCCc
    sgVGLUT2 (S): caccgAGAGGACGGTAAGCCCCTGG
    sgVGLUT2 (AS): aaacCCAGGGGCTTACCGTCCTCTc

Lentiviruses were produced by transfection of viral HEK-293T cells with SV-tag in combination with the VSV-G envelope and CMV ΔVPR packaging plasmids. Twenty-four hours after transfection, the media was changed to fresh DMEM (Invitrogen) with 20% heat inactivated fetal bovine serum (Gemini BioProducts). Forty-eight hours after transfection, the virus containing supernatant was collected from the cells and centrifuged at 1000 g for 5 min to remove cells and debris. Supernatants were stored for up to 1 week at 4°C and added to plated cortical neurons at DIV 3.

## SV isolation from cortical cultures

Five million neurons plated on a 10 cm tissue culture dish at DIV12-14 were used for each immunoisolation. No more than two plates were processed at a time to increase the speed of isolation. All buffers and tubes used were prechilled on ice, with the exception of the metabolite extraction mix, which was kept on dry ice, and all steps were performed swiftly. Neurons were placed on ice to chill, rapidly rinsed with ice-cold phosphate buffered saline (PBS) to remove residual media, and gently scraped into 1 ml PBS. Cells were pelleted by a brief centrifugation step at 2400 g for 40 s. The PBS was aspirated and 1 ml of homogenization buffer (320 mM sucrose, 4 mM HEPES NaOH, pH 7.4) supplemented with cOmplete EDTA-free protease inhibitor (Roche) and 1 mM ATP NaOH, pH 7.4. The cell pellet was uniformly resuspended with a 1 ml large bore tip (Fisher Scientific) and transferred to a 2 ml homogenizer (VWR International). To generate synaptosomes, the cells were homogenized with 25 steady strokes, with care taken to minimize formation of air bubbles. The homogenate was centrifuged at 2400 g for 40 s to pellet unbroken cells. The supernatant was then transferred to a new tube and centrifuged at 14000 g for 3 min to pellet synaptosomes. The supernatant was carefully removed and the pellet resuspended in 100 µl of homogenization buffer using a 200 ul large bore tip (Fisher Scientific). 900 µl of ice-cold ddH$_2$O (MS grade) was added, and the liquid was immediately transferred to a 2 ml homogenizer. To lyse synaptosomes, the cells were homogenized with 12 steady strokes. Osmolarity was restored following homogenization by the addition of 1 mM ATP, 5 mM HEPES NaOH and 1X cOmplete EDTA-free protease inhibitor (final osmolality = 70 Osm). Finally, the homogenate was centrifuged at 17000 g for 3 min to remove any unbroken synaptosomes and debris. This process takes a total of ~12 min.

To immunoisolate SVs, 150 µl of suspended, prewashed magnetic HA beads (Thermo Fisher Scientific) were added to the supernatant and incubated at 4℃ with end-over-end rotation for 15 min. For washes, beads were captured with a DynaMag Spin Magnet (Thermo Fisher Scientific) for 40 s. Four washes were performed in succession by the addition of 1 ml of KPBS ( 136 mM KCl, 10 mM KH2PO4, pH 7.25 in MS grade water) (*Chen et al., 2016*). Following the final wash, 25% of the KPBS-bead suspension was aliquoted for immunoblot analysis and the remaining 75% was subjected to metabolite extraction with 100 µl of 80% methanol/20% water supplemented with 500 µM internal amino acid standards. To ensure complete extraction, the beads were incubated with extraction mix for at least 10 min on dry ice prior to being separated from the mix. Extracted metabolites were subjected to a final 17000 g spin for 3 min to remove any particulates and stored at −80℃ until the MS run. Results reported are from a single run with each condition containing at least three technical replicates.

## Immunoblotting

Protein from lysates were denatured by the addition of 50 µl of sample buffer. For whole-cell lysates, 0.5 µl of Benzonase (EMD Millipore) was added and incubated with the lysates for at least 5 min to break down genomic DNA. Samples were resolved by 8–16% SDS-PAGE, transferred for 2 hr at room temperature at 45 V to 0.45 mm PVDF membranes, and analyzed by immunoblotting as described previously (*Chantranupong et al., 2016*). Briefly, membranes were blocked with 5% milk prepared in TBST (Tris-buffered Saline with Tween 20) for at least 5 min at room temperature, then incubated with primary antibodies in 5% BSA TBST overnight at 4℃ with end-over-end rotation. Primary antibodies targeting the following proteins were used at the indicated dilutions and obtained from the denoted companies: synaptophysin 1:2000 (SySy Cat # 101002), SV2A 1:2000 (SySy # 119003), HA 1:1000 (CST #C29F4), NMDAR 1:1000 (SySy #114011) (1:1000), synaptobrevin 1:5000 (SySy #104211), calreticulin 1:300 (CST #12238), VDAC 1:200 (CST#4661), LC3B 1:300 (CST #2775), GAPDH 1:2000 (CST #2118), synaptotagmin 1:1000 (SySy #105011), VGLUT1 1:2000 (SySy #135303), and VGLUT2 1:500 (SySy #135421). Following overnight incubation, membranes were washed three times, 5 min each, with TBST and incubated with the corresponding secondary antibodies in 5% milk (1:5000) for 1 hr at room temperature. Membranes were then washed three more times, 5 min each, with TBST before being visualized using enhanced chemiluminescence (Thermo Fisher Scientific).

## Endogenous tagging of synaptophysin

An endogenous triple HA-tag was appended onto the C-terminus of synaptophysin using the vSLENDR method (*Nishiyama et al., 2017*). An AAV construct (below) containing a guide targeting

the C-terminus of synaptophysin (bracketed text), gRNA scaffold ( italic text), 5' and 3' homology arms flanking this region (normal text), and the triple HA-tag ( text within parentheses) replaced the mEGFP-cmak2a HDR cassette in the backbone of the pAAV-HDR-mEGFP-camk2a (Addgene #104589). This construct, which we term pAAV-HDR-sphys-3XHA, along with pAAV-EFS-SpCas9 were packaged at Boston Children's Viral Core. Cortical neurons cultured on a cover slip were coinfected on DIV3 with 3.45e$^7$genome copies (GC) of AAV-EFS-SpCas9 and 1.42e$^{10}$ GC of AAV-HDR-sphys-3XHA. Neurons were processed for immunostaining at DIV12-14.

5' [ TTCTCCAATCAGATGTAAT**C**]*GTTTTAGAGCTAGAAATAGCAAGTTAAAATAAGGCTAGTCCGTTATCAACTTGAAAAAGTGGCACCGAGTCGGTGCTTTTTTGAATTC*TTTTGGTTTTGTTTGA-GACAGGATCTACTTATGTGACCCTGGCTGTCCTGGAACTCACTACTCAGACCAGACTGGCCTCAAACTCACAGACCTCTGCTTGCCTCTGCCTCCTGAGTACTAAGATGAAGACTGCACCACCA-CACCCAGCCCAAAAATGAGTTGTTTGAGGCTGACTTTCATGTTGCACAGGCTAGCCTCAAACTATGAATTTAAAGGTAGACTTGAATTTCTGGGTAGTGGAGGCAGAGACAGGCGACTTCTATGAGTTCCAGGCCAGCCTGGTCTACAGAGTGAATTCCAGGACAGCCAGGGCTGCAGAGAGACCCTGTCTCAAAAAAAAAAAGCTAGCCTTGAAGTGATCGCCCCTGCCTCCAGCTTCCTAAGATTACAAGATGTGGGCCTTCAGACTTGTCCATGTAAACACTGATAGAAGTTGAACATCATGGGAATCTAACACACA-CACACACACTCCCAAGTTTTTCTGTACACTGATAGTCATAGAGGCCCACGAATTTATGCCCTAAAAATGCCCATTCCTGTTCACTCAGCCTCAAAGACCCTGGGGCTGCCGAGGCAATGGGTAA-GAGACAACAGCTTTGGTCATGTCTCCCTGCAGGTGTTTGGCTTCCTGAACCTGGTGCTCTGGGTTGGCAACCTATGGTTCGTGTTCAAGGAGACAGGCTGGGCCGCCCCATTCATGCGCGCACCTCCAGGCGCCCCAGAAAAGCAACCAGCTCCTGGCGATGCCTACGGCGATGCGGGCTATGGGCAGGGCCCCGGAGGCTATGGGCCCCAGGACTCCTACGGGCCTCAGGGTGGTTATCAACCCGATTACGGGCAGCCAGCCAGCGGCGGTGGCGGTGGCTACGGGCCTCAGGGCGACTATGGGCAGCAAGGCTACGGCCAACAGGGTGCGCCCACCTCCTTCTCCAACCAAATG(GGAGG-GAGCGGCTATCCCTATGACGTGCCTGATTACGCCGGCACAGGATCCTACCCCTATGATGTGCCTGACTACGCTGGCAGCGCCGGATACCCTTATGATGTGCCTGATTATGCTTAA)TCTGGTGAGTGA-CAACTGGGCGGATGCGGTAGGCAGGGAGCATACAAGGAGTGAAGTTTGAAGGAACCAATAGATAGGCAGAACCAAAAAAAAAAAAAAGGTGAACTTGGTAAAACTAGCCAATGAGAGGAACCGTGA-GAAGGAAGGGGACGGAGCAGTGCTCAGAGTAACCAATGAAAGGAGTGTAGGGGCACTTGCGCAGTGGAGAATCACCAAAGTGGTGTAGGTTTCCAGGAAGGGAAGGGGAGGAGGGTCTTTGAAATCATTGGTAAACCAATAGGCGGTAACGCCAGTAGGTGGAAGAAGGTAAACACGTTGGGTTTTGAAGGGCGCTAGCGCTAAAGCAGGATGTAGGTCAGCTGCTACCTCTCCTTAACCCTTTAATGAAAGAGAGAGTTTGGAATTTCAAATGAGGAAAAGGGGAGGGCTGGAGGCCTTA-GAAACACGAGTATGCCTTTTTGTTGGGCCTTTAAAAAATGAATGCCGCCGGACGGTGGAGGCG-CACGCCTTTAATCCCAGCACTTGGGAGGCAGAAGCAGGCGGAGTTTTTGAGTTCGAGGCCAGCCTGGTCTACAAAGTGAGTTCCAGGACAGCCAGGGCTATACAGAGAAACCCTGTCGCGAAAAAAAAAAAAAAAAAAAAAAACCCTGCCTGGTGTGATGGAGCACATATATAATCCCAG-CACTTGAGAGGTAGAGGCATGGGGATTGCAAGTTCTCGAGTCCTGCGTGGTCAGTATAGCCCAATCCTGTCTTAAACAGAGACGGTAACAGCATCTAGGTGGGAGCAGATGTGGTCCTGGGTGAGCCTTCTACAGCAACCCACATTTAATTGTTTTTAAACTCCTTGGACAGGCTCTGAGACACACCTTTAAG-CACAGCTCTGGGGGAATTAGAGACAGGCCTAGGTCTCTTGTTTTGCAAAGCAATTTCCAGGCTGC 3'.

## Immunohistochemistry

Cells were fixed in 4% PFA 4% sucrose in PBS for 10 min, and washed in PBS twice for 10 min each with shaking at room temperature. Cells were then blocked in 1 ml of blocking solution (10% BSA, 10% normal goat serum, PBS) for 30 min at room temperature. Cells were washed with 1 ml of TBST (0.2% TritonX-100 in PBS) for 10 min. Primary antibodies were suspended in BTBST (1% BSA, 1% normal goat serum PBS) and 300 ul was added to each slip. The following concentrations were used: synapsin 1:500 (CST #5297), synaptophysin 1:1000 (SySy 101004), synaptotagmin 1:1000, and HA 1:500 (CST#2367). Cells were incubated overnight at 4°C or at room temperature for 1.5 hr. Following this, cells were washed three times, 10 min each, in 1 ml of PBS with rocking at room temperature. The following secondary antibodies (Thermo Fisher Scientific) were diluted in BTBST and added at a 1:500 dilution: goat anti-rabbit Alexa Fluor 488, goat anti-guinea pig Alexa Fluor 488, goat anti-rabbit Alexa Fluor 647. Cells were covered from light and incubated at room temperature for 2 hr, washed three more times in PBS for 10 min each and mounted on cover slides with

Floromount G (Thermo Fisher Scientific). Slips were imaged an Olympus VS120 slide scanning microscope.

## Transmission electron microscopy

To free SVs from beads, we relied on a protease strategy as the HA binding strength to its cognate antibody is too strong to be dissociated by peptide competition. We used ficin (Sigma), a cysteine protease that can rapidly digest murine monoclonal IgGs (*Mariani et al., 1991*). Following the IP, SVs were equilibrated with three washes of ficin buffer (50 mM Tris pH 7.0, 2 mM EDTA, 1 mM cysteine). The supernatant was removed and replaced with 100 µl of ficin buffer supplemented with 5 mg/ml ficin (Sigma). SVs were incubated at 37°C for 20 min to enable ficin to be active. Supernatant was removed and immediately chilled on ice prior to EM analysis. Images were acquired at the Electron Microscopy Core at Harvard Medical School. 5 µl of the sample was adsorbed for 1 min to a carbon coated grid that had been made hydrophilic by a 30 s exposure to a glow discharge. Excess liquid was removed with a filter paper (Whatman #1) and the samples were stained with 0.75% uranyl formate for 30 s. After removing the excess uranyl formate with a filter paper the grids were examined in a JEOL 1200EX Transmission electron microscope or a TecnaiG$^2$ Spirit BioTWIN and images were recorded with an AMT 2 k CCD camera.

## Quantitative immunoblotting

To determine the protein concentration of the whole-cell input, 50 µl of the initial cellular lysate was pelleted with a 17,000 g spin at 4°C for 5 min and the whole-cell pellet lysed for 10 min in Triton elution buffer (1% TritonX-100, 500 mM NaCl, 5 mM HEPES pH 7.4, 1X protease inhibitors). The lysate was clarified with at 17,000 g spin at 4°C for 10 min and the supernatant analyzed for protein content with the BCA assay (Pierce). To determine the protein concentration of isolated SVs, the final HA immunoisolate from SV-tag expressing cells was eluted in 70 µl Triton buffer for 15 min at 32°C and 30 µl was analyzed for protein content with the BCA assay. A fraction of the immunoisolate and the whole-cell input were loaded onto an SDS-PAGE gel alongside varying concentrations of recombinant His tagged synaptobrevin (Abcam) and immunoblotting was performed for synaptobrevin (Synaptic Systems). The immunoblot signal intensity was quantified via ImageJ and the concentration of synaptobrevin extrapolated based on the standard curve generated by recombinant synaptobrevin.

## Luminescence assay for glutamate

Following immunoisolation, SVs were permeabilized with 70 µl of Triton elution buffer and incubated at 37°C for 20 min to ensure complete permeabilization. 15 µl of this eluent was saved for immunoblot analyses. To detect glutamate, the Glutamate Glo Assay Kit (Promega) was used. 50 µl of eluent was combined into a 96-well plate with a 50 µl mix of Luciferin detection solution, which contains reductase, reductase substrate, glutamate hydrogenase, and NAD as specified. The mixture was incubated at room temperature for 1 hr and luminescence was detected with a florescent plate reader (BioTek).

## Proteomics run and analysis

To ensure sufficient yields for proteomic analysis, each sample combined cells from three plates, for a total of 15 million neurons. Following immunoisolation, SVs were permeabilized with 70 µl of Triton elution buffer at 32°C for 20 min. Eluents from the three plates were pooled into a common tube. 20 µl of this mix was saved for immunoblot analysis. The remaining eluent was transferred to a new tube and subjected to TCA precipitation. Briefly, the volume of eluent was raised to 400 µl with MS grade, ice-cold water. 100 µl of 100% tricloroacetic acid (TCA) was added to this mixture.

Precipitated proteins were submitted to the Taplin Mass Spectrometry Core for proteomic analysis. There, samples were digested with 50 µl of 50 mM ammonium bicarbonate solution containing 5 ng/µl modified sequencing-grade trypsin (Promega, Madison, WI) at 4°C. After 45 min., the excess trypsin solution was removed and replaced with 50 mM ammonium bicarbonate solution to just cover the gel pieces. Samples were then placed in a 37°C room overnight. Peptides were later extracted by removing the ammonium bicarbonate solution, followed by one wash with a solution containing 50% acetonitrile and 1% formic acid. The extracts were then dried in a speed-vac (~1 hr). The samples were then stored at 4°C until analysis.

On the day of analysis the samples were reconstituted in 5–10 µl of HPLC solvent A (2.5% aceto-nitrile, 0.1% formic acid). A nano-scale reverse-phase HPLC capillary column was created by packing 2.6 µm C18 spherical silica beads into a fused silica capillary (100 µm inner diameter x ~ 30 cm length) with a flame-drawn tip (*Peng and Gygi, 2001*). After equilibrating the column each sample was loaded via a Famos auto sampler (LC Packings, San Francisco CA) onto the column. A gradient was formed and peptides were eluted with increasing concentrations of solvent B (97.5% aceto-nitrile, 0.1% formic acid).

As peptides eluted they were subjected to electrospray ionization and then entered into an LTQ Orbitrap Velos Pro ion-trap mass spectrometer (Thermo Fisher Scientific, Waltham, MA). Peptides were detected, isolated, and fragmented to produce a tandem mass spectrum of specific fragment ions for each peptide. Peptide sequences (and hence protein identity) were determined by matching protein databases with the acquired fragmentation pattern by the software program, Sequest (Thermo Fisher Scientific, Waltham, MA) (*Eng et al., 1994*). All databases include a reversed version of all the sequences and the data was filtered to between a one and two percent peptide false dis-covery rate. Results reported are from a single run with each condition containing three technical replicates.

## Mice

The following mouse strains/lines were used in this study: CD-1 IGS (Charles River Laboratories, Stock # 022); C57BL/6J (The Jackson Laboratory, Stock # 000664); DAT-IRES-Cre (The Jackson Labo-ratory, Stock #006660)(referred to as *Slc6a3*IRES-Cre mice); VGAT-IRES-Cre (The Jackson Laboratory, Stock #016962) (referred to as *Slc32a1*Cre mice); genetically targeted *Adora2a*Cre BAC transgenic mice (GENSAT, founder line KG139), which express Cre under transcriptional control of the adeno-sine A2A receptor genomic promoter (*Durieux et al., 2009*). All animals were kept on a regular 12:12 light/dark cycle under standard housing conditions. All experimental manipulations were per-formed in accordance with protocols approved by the Harvard Standing Committee on Animal Care following guidelines described in the US National Institutes of Health Guide for the Care and Use of Laboratory Animals.

## SV isolation from whole brain

Mice were rapidly anesthetized with isoflurane and brains were quickly extracted from mice on ice. To ensure more efficient homogenization, each brain was divided in half along the midline and trans-ferred immediately to a homogenizer containing 1.5 ml of ice-cold lysis buffer (KPBS supplemented with 1 mM ATP and cOmplete EDTA-free protease inhibitor). Brains were rapidly lysed with 30 strokes using a motorized homogenizer, taking care not to introduce bubbles. Lysates were trans-ferred to prechilled 2 ml tubes and centrifuged at 17, 000 g for 3 min to pellet unbroken cells, con-taminating organelles and debris. Supernatants were transferred to new 1.5 ml tubes and subjected to IP with 150 µl HA beads for 15 min with end-over-end rotation. Following the IP, the beads were washed four times in KPBS supplemented with 500 mM NaCl to enhance cleanliness. In the final wash, the IP was incubated with end-over-end rotation at 4°C for 5 min to further remove contami-nants. Following this final wash, immunoprecipitates were processed for immunoblot and metabolite analysis as described for cortical culture SVs described above.

## Stereotaxic injections

Injections were performed as previously described (*Huang et al., 2019*). AAVs for SV-tag was obtained from Boston Children's Virus Core and infused at a concetration of ~1012 GC/ml. AAVs were infused into target regions at approximately 50 nl/min using a syringe pump (Harvard Appara-tus, #883015), and pipettes were slowly withdrawn (<10 µm/s) at least 8 min after the end of the infusion. All coordinates are relative to Bregma along the anterior-posterior (AP) axis and medial-lat-eral (ML) axis, and relative to the pial surface along the dorsoventral axis (DV). Coordinates for injec-tions are as follows: cortex site 1: AP = −2.5 mm, ML = −2.0 mm, DV = −0.4 mm, cortex site 2: AP = −1.0 mm, ML = −1.5 mm, DV = −0.4 mm, striatum: AP = +0.6 mm, ML = 1.7 mm, DV = −3.33 mm, hippocampus AP = −2.5 mm, ML = −1.5 mm, DV = −1.5 mm; hippocampus transverse (for paired-pulse ratio measurements): AP = −2.9 mm, ML = −3.15 mm, DV = −3.3 mm, substantia nigra: AP = −3.3 mm, ML = −1.5 mm, DV = −4.3 mm. Bilateral injections were performed for all

mice used, with the exception of cortex, in which four injections were performed to maximally cover the cortical area. Following wound closure, mice were placed in a cage with a heating pad until their activity was recovered before returning to their home cage. Mice were given pre- and post-operative oral carprofen (MediGel CPF, 5 mg/kg/day) as an analgesic, and monitored daily for at least 4 days post-surgery.

## Immunohistochemistry

Mice were anaesthetized by isoflurane inhalation and perfused cardiacly with PBS followed by 4% PFA in PBS. Brains were extracted and stored in 4% PFA PBS for at least 8 hr. Brains were sliced into 70 μm thick free-floating sections with a Leica VT1000 s vibratome. Selected slices were transferred to a clean six well plate and rinsed three times, 5 min each in PBS. They were then blocked with rotation at room temperature for an hour in blocking buffer (5% normal goat serum (Abcam), 0.2% TritonX-100 PBS). Blocking buffer was removed and replaced with 500 μl of a solution containing a 1:500 dilution of anti- tyrosine hydroxylase antibody (Millipore Sigma). Slices were incubated overnight with side to side rotation at 4°C. The next day, slices were transferred to a clean well and washed five times, 5 min each in PBST (PBS with 0.2% TritonX-100). Following the final wash, slices were incubated for 1.5 hr in 500 μl of secondary antibody (goat anti mouse Alexa Fluor 647) diluted 1:500 in blocking buffer. Slices were washed four times in PBST, then four times in PBS for 5 min (5 min for each wash) before mounting with Floromount G (Thermo Fisher Scientific). Slices were imaged an Olympus VS120 slide scanning microscope, including those housed in the Neuro Imaging Facility.

## Electrophysiology

Coverslips containing cultured neurons or acute brain slices were transferred into a recording chamber mounted on an upright microscope (Olympus BX51WI) and continuously superfused (2–3 ml min$^{-1}$) with ACSF containing (in mM) 125 NaCl, 2.5 KCl, 25 NaHCO$_3$, 2 CaCl$_2$, 1 MgCl$_2$, 1.25 NaH$_2$PO$_{4,}$ and 25 glucose (295 mOsm kg$^{-1}$). ACSF was warmed to 32–34°C by passing it through a feedback-controlled in-line heater (SH-27B; Warner Instruments). Cells were visualized through a 60X water-immersion objective with either infrared differential interference contrast optics or epifluorescence to identify tdTomato$^+$ cells. For whole-cell voltage clamp recordings, patch pipettes (2–4 MΩ) pulled from borosilicate glass (G150F-3, Warner Instruments) were filled with a Cs$^+$-based low Cl$^-$ internal solution containing (in mM) 135 CsMeSO$_3$, 10 HEPES, 1 EGTA, 3.3 QX-314 (Cl$^-$ salt), 4 Mg-ATP, 0.3 Na-GTP, 8 Na$_2$-phosphocreatine (pH 7.3 adjusted with CsOH; 295 mOsm·kg$^{-1}$) For voltage clamp recordings, mEPSCs were recorded for 5 min in the presence of 1 μM tetrodotoxin (Tocris), 10 μM CPP (Tocris), and 10 μM gabazine (Tocris) at a holding potential of −70 mV. Paired evoked EPSCs for probability of release measurements were recorded as previously described (*Jackman et al., 2016*). Briefly, a cut was made between CA3 and CA1 to prevent recurrent excitation. Extracellular stimulation was performed with a stimulus isolation unit (Iso-flex). Bipolar electrodes (PlasticOne) were placed near CA3 proximal to the cut and stimulation parameters were 20 Hz, 50–100 μA. Paired evoked-EPSCs from CA1 cells were recorded at a holding potential of −70 mV with 10 μM gabazine added to the bath.

## Acute brain slice preparation

Brain slices were obtained from 2- to 4-month-old mice (both male and female) using standard techniques. Mice were anaesthetized by isoflurane inhalation and perfused cardiacly with ice-cold ACSF containing (in mM) 125 NaCl, 2.5 KCl, 25 NaHCO$_3$, 2 CaCl$_2$, 1 MgCl$_2$, 1.25 NaH$_2$PO$_4$, and 25 glucose (295 mOsm kg$^{-1}$). Brains were blocked and transferred into a slicing chamber containing ice-cold ACSF. Coronal slices of striatum for amperometric recordings or transverse slices of hippocampus (for probability of release measurements) were cut at 300 μm thickness with a Leica VT1000 s vibratome in ice-cold ACSF, transferred for 10 min to a holding chamber containing choline-based solution (consisting of (in mM): 110 choline chloride, 25 NaHCO$_3$, 2.5 KCl, 7 MgCl$_2$, 0.5 CaCl$_2$, 1.25 NaH$_2$PO$_4$, 25 glucose, 11.6 ascorbic acid, and 3.1 pyruvic acid) at 34°C then transferred to a secondary holding chamber containing ACSF at 34C for 10 min and subsequently maintained at room temperature (20–22°C) until use. All recordings were obtained within 4 hr of slicing. Both choline solution and ACSF were constantly bubbled with 95% O$_2$/5% CO$_2$.

## Amperometric recordings

To deplete presynaptic terminals of dopamine, $Slc6a3^{IRES-Cre}$ mice were administered the VMAT2 antagonist tetrabenazine (Sigma, 30 mg kg$^{-1}$ intraperitoneally) 2 hr before slicing. Control mice were injected with a DMSO/saline mixture containing the same proportion of both solvents as would be used for a tetrabenazine injection. Constant-potential amperometry was performed as previously described (*Tritsch et al., 2012*). Briefly, glass-encased carbon-fiber microelectrodes (CFE1011 from Kation scientific - 7 μm diameter, 100 μm length) were placed approximately 50–100 μm within dorsal striatum slices and held at a constant voltage of + 600 mV vs. Ag/AgCl by a Multiclamp 700B amplifier (Molecular Devices). Electrodes were calibrated with fresh 5 μM dopamine standards in ACSF to determine CFE sensitivity and to allow conversion of current amplitude to extracellular dopamine concentration. Dopaminergic terminals surrounding the CFE were stimulated by Bipolar electrodes with 0.1 ms and 100–250 μA delivered at 3 min intervals.

## Data acquisition and analysis

Membrane currents were amplified and low-pass filtered at 3 kHz using a Multiclamp 700B amplifier (Molecular Devices), digitized at 10 kHz and acquired using National Instruments acquisition boards and a custom software (https://github.com/bernardosabatini/SabalabAcq) written in MATLAB (Mathworks). Amperometry and electrophysiology were analyzed offline using Igor Pro (Wavemetrics). Detection threshold for mEPSCs was set at 7 pA. Averaged waveforms were used to obtain current latency, peak amplitude, 10–90% rise time and decay time. Current onset was measured using a threshold set at three standard deviations of baseline noise. Peak amplitudes were calculated by averaging over a 2 ms window around the peak. Data were compared statistically by unpaired two-tailed Student's t-test. p values less than 0.05 were considered statistically significant.

## GC/MS

GC-MS analysis was carried out and analyzed as described (*Parker et al., 2017*). In brief, dried, extracted metabolites were derivatized using a MOX-tBDMCS method and analyzed by GC-MS using a DB-35MS column (30 m × 0.25 mm i.d., 0.25 μm) in an Agilent 7890B gas chromatograph interfaced with a 5977B mass spectrometer. Metabolites were identified by unique fragments and retention time in comparison to known standards. Peaks were picked in OpenChrom and integrated and corrected for natural isotopic abundance using in-house algorithms adapted from *Fernandez et al., 1996*; *Lewis et al., 2014*; *Wenig and Odermatt, 2010*.

## LC-MS/MS with the hybrid metabolomics method

Samples were subjected to an LCMS analysis to detect and quantify known peaks. A metabolite extraction was carried out on each sample with a previously described method (*Pacold et al., 2016*). The LC column was a Millipore ZIC-pHILIC (2.1 × 150 mm, 5 μm) coupled to a Dionex Ultimate 3000 system and the column oven temperature was set to 25°C for the gradient elution. A flow rate of 100 μl/min was used with the following buffers; (A) 10 mM ammonium carbonate in water, pH 9.0, and (B) neat acetonitrile. The gradient profile was as follows; 80–20%B (0–30 min), 20–80%B (30–31 min), 80–80%B (31–42 min). Injection volume was set to 1 μl for all analyses (42 min total run time per injection). MS analyses were carried out by coupling the LC system to a Thermo Q Exactive HF mass spectrometer operating in heated electrospray ionization mode (HESI). Method duration was 30 min with a polarity-switching data-dependent Top five method for both positive and negative modes. Spray voltage for both positive and negative modes was 3.5kV and capillary temperature was set to 320°C with a sheath gas rate of 35, aux gas of 10, and max spray current of 100 μA. The full MS scan for both polarities utilized 120,000 resolution with an AGC target of 3e6 and a maximum IT of 100 ms, and the scan range was from 67 to 1000 $m/z$. Tandem MS spectra for both positive and negative mode used a resolution of 15,000, AGC target of 1e5, maximum IT of 50 ms, isolation window of 0.4 m/z, isolation offset of 0.1 m/z, fixed first mass of 50 m/z, and 3- way multiplexed normalized collision energies (nCE) of 10, 35, 80. The minimum AGC target was 1e4 with an intensity threshold of 2e5. All data were acquired in profile mode.

## Metabolomics data processing

### Relative quantification of metabolites

The resulting Thermo RAW files were converted to mzXML format using ReAdW.exe version 4.3.1 to enable peak detection and quantification. The centroided data were searched using an in-house python script Mighty_skeleton version 0.0.2 and peak heights were extracted from the mzXML files based on a previously established library of metabolite retention times and accurate masses adapted from the Whitehead Institute (*Chen et al., 2016*), and verified with authentic standards and/or high-resolution MS/MS spectral manually curated against the NIST14MS/MS (*Voge et al., 2016*) and METLIN (2017) (*Smith et al., 2005*) tandem mass spectral libraries. Metabolite peaks were extracted based on the theoretical $m/z$ of the expected ion type for example $[M+H]^+$, with a ± 5 part-per-million (ppm) tolerance, and a ± 7.5 s peak apex retention time tolerance within an initial retention time search window of ±0.5 min across the study samples.

### Detection of untargeted features

The MS1 level data in both positive and negative mode were searched for representative features across all study files using an in-house python script called Ungrid (version 0.5). The algorithm reduces all detected MS1 peak data into representative features by sorting intensity from high to low (across all samples) and then applying an array bisection algorithm (python v3.0.1) on the m/z and retention time values with custom tolerances (25 ppm m/z tolerance for peak discrimination, 0.5 min RT delta for chromatographic discrimination, 1e5 minimum intensity, 10X signal to noise (within spectrum)). The output is a list of representative high intensity features of a defined m/z and retention time. These feature intensities were then extracted across all samples using Mighty_skeleton (above) to give the peak intensities for each feature in each sample.

### Metabolomics informatics

The resulting data matrices of metabolite intensities for all samples and blank controls (either retention time library data or untargeted data) was processed with an in-house statistical pipeline Metabolyze version 1.0 and final peak detection was calculated based on a signal to noise ratio (S/N) of 3X compared to blank controls, with a floor of 10,000 (arbitrary units). For samples where the peak intensity was lower than the blank threshold, metabolites were annotated as not detected, and the threshold value was imputed for any statistical comparisons to enable an estimate of the fold change as applicable. The resulting blank corrected data matrix was then used for all group-wise comparisons, and t-tests were performed with the Python SciPy (1.1.0) (*Jones et al., 2020*) library to test for differences and generate statistics for downstream analyses. Any metabolite with p-value<0.05 was considered significantly regulated (up or down). Any outliers were omitted with the Grubb's outlier test. Values are reported as $\log_2$ fold changes ± standard error of the mean. In order to adjust for significant covariate effects (as applicable) in the experimental design the R package, DESeq2 (1.24.0) (*Love et al., 2014*) was used to test for significant differences. Data processing for this correction required the blank corrected matrix to be imputed with zeroes for non-detected values instead of the blank threshold to avoid false positives. This corrected matrix was then analyzed utilizing DESeq2 to calculate the adjusted p-value in the covariate model.

## Targeted LC/MS for absolute quantification of dopamine

Metabolite profiling was conducted on a QExactive bench top orbitrap mass spectrometer equipped with an Ion Max source and a HESI II probe, which was coupled to a Dionex UltiMate 3000 HPLC system (Thermo Fisher Scientific, San Jose, CA). External mass calibration was performed using the standard calibration mixture every 7 days. 5 μl were injected onto a SeQuant ZIC-pHILIC 150 × 2.1 mm analytical column equipped with a 2.1 × 20 mm guard column (both 5 mm particle size; EMD Millipore). The following method was adapted from *Tufi et al., 2015*: Buffer A was 10 mM ammonium formate with 0.2% formic acid in 90% acetonitrile; Buffer B was water with 0.2% formic acid. The column oven and autosampler tray were held at 25°C and 4°C, respectively. The chromatographic gradient was run at a flow rate of 0.300 ml/min as follows: 0–2 min: hold at 0% B; 2.5–15 min.: linear gradient form 0–40% B; 15.5–16 min.: linear gradient from 40–0% B; 16–21 min.: hold at 0% B. The mass spectrometer was operated in full-scan, polarity-switching mode, with the spray voltage set to 3.0 kV, the heated capillary held at 275°C, and the HESI probe held at 350°C. The

sheath gas flow was set to 40 units, the auxiliary gas flow was set to 15 units, and the sweep gas flow was set to one unit. MS data acquisition was performed in a range of $m/z$ = 70–1000, with the resolution set at 70,000, the AGC targeted at $1 \times 10^6$, and the maximum injection time was 20 msec. In addition, timed targeted selected ion monitoring (tSIM) scans were included in positive mode to enhance detection of Dopamine ($m/z$ 154.08626). MS settings were as described above, with the AGC target set at $1 \times 10^5$, the maximum injection time was 200 ms, and the isolation window was 1.0 $m/z$. Raw peak areas were normalized to phenylalanine-13C9-15N as an internal standard.

## Acknowledgements

We thank all members of the Sabatini lab for helpful suggestions and advice, in particular T Consedine and G Radeljic for technical assistance and K Mastro, S Melzer, and A Girasole for manuscript input. We also thank WW Chen and MA Remaileh for insightful input in methodology and/or data analysis. We thank the Electron Microscopy Core at Harvard Medical School for generous assistance with the EM imaging, CA Lewis at the Whitehead Metabolite Profiling Core Facility for the MS runs and analysis of dopamine, and the Taplin Mass Spectrometry Core for proteomic runs and analysis. This work was supported by grants from the Howard Hughes Medical Institute and R37NS046579 from NINDS to BLS; the Hanna Gray Fellowship from the Howard Hughes Medical Institute to LC; the Mary Kay Foundation (Cancer Research Grant 017–032), the Shifrin-Myers Breast Cancer Discovery Fund at NYULMC, a V Foundation V Scholar Grant funded by the Hearst Foundation (V2017-004), and an NCI K22 Career Transition Award (1K22CA212059) to MEP; (NIH) R01 NS108151-01, (FNIH) RFA 2018-PACT001, and (NIH) HHS-NIH-NIAD-BAA2018 to D.R.J.; and NINDS P30 Core Center Grant #NS072030 to the Neuro Imaging Facility.

## Additional information

### Funding

| Funder | Grant reference number | Author |
| --- | --- | --- |
| Howard Hughes Medical Institute | Investigator | Bernardo L Sabatini |
| National Institute of Neurological Disorders and Stroke | R37NS046579 | Bernardo L Sabatini |
| Howard Hughes Medical Institute | Hanna Gray Fellowship | Lynne Chantranupong |
| Mary Kay Foundation | Cancer Research Grant 017-032 | Michael E Pacold |
| Hearst Foundations | V Foundation V Scholar Grant (V2017-004) | Michael E Pacold |
| National Cancer Institute | K22 Career Transition Award (1K22CA212059) | Michael E Pacold |
| NIH | R01 NS108151-01 | Drew R Jones |
| FNIH | RFA 2018-PACT001 | Drew R Jones |
| NIH | HHS-NIH-NIAD-BAA2018 | Drew R Jones |

The funders had no role in study design, data collection and interpretation, or the decision to submit the work for publication.

### Author contributions

Lynne Chantranupong, Conceptualization, Resources, Data curation, Formal analysis, Supervision, Funding acquisition, Validation, Investigation, Visualization, Methodology, Writing - original draft, Writing - review and editing; Jessica L Saulnier, Investigation, Writing - review and editing; Wengang Wang, Investigation, Methodology, Writing - review and editing; Drew R Jones, Michael E Pacold, Resources, Data curation, Formal analysis, Investigation, Methodology, Writing - review and editing;

Bernardo L Sabatini, Conceptualization, Resources, Data curation, Formal analysis, Supervision, Funding acquisition, Investigation, Visualization, Methodology, Writing - review and editing

## Author ORCIDs
Lynne Chantranupong (iD) https://orcid.org/0000-0001-9814-5264
Michael E Pacold (iD) http://orcid.org/0000-0003-3688-2378
Bernardo L Sabatini (iD) https://orcid.org/0000-0003-0095-9177

## Ethics
Animal experimentation: All experimental manipulations were performed in accordance with protocols (#03551) approved by the Harvard Standing Committee on Animal Care following guidelines described in the US National Institutes of Health Guide for the Care and Use of Laboratory Animals.

## Decision letter and Author response
Decision letter https://doi.org/10.7554/eLife.59699.sa1
Author response https://doi.org/10.7554/eLife.59699.sa2

## Additional files

### Supplementary files
• Supplementary file 1. Proteomics profile of immunoprecipitates from primary cortical neurons expressing SV-tag vs. uninfected neurons, related to Figure S1D.

• Supplementary file 2. List of the metabolites interrogated in the targeted LC/MS runs, related to *Figure 2C,D and F*, *Figure 4D,E and G*, and Figure S4H.

• Supplementary file 3. Metabolites detected in SVs from cortical cultures via GC/MS and LC/MS and their associated calculations of fold changes, significance, and SEM, related to *Figure 2*.

• Supplementary file 4. Metabolites detected via global LC/MS performed on cortical culture SVs, treated with DMSO or BafilomycinA, related to *Figure 3*.

• Supplementary file 5. Metabolites detected in SVs from hippocampus and striatum via LC/MS and their associated calculations of fold changes, significance, and SEM, related to *Figure 4*.

• Transparent reporting form

### Data availability
All data generated or analysed during this study are included in the manuscript and supporting files. Source data files have been provided for all figures.

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
