## [Decision Letter]

**Acceptance summary:**

The manuscript from Chantranupong et al. reports a new method for the immune-isolation of SVs based on the expression of a synaptophysin variant containing a HA-tag. This method allows for rapid isolation of SVs with high purity from both neuronal culture and mouse brain. The authors then carried out metabolomics profiling using LC/MS, allowing for detection, among many other metabolites, of neurotransmitters. This study addresses the classical question of SV content that has only become more important in recent years, due in particular to the accumulating evidence for co-release of classical neurotransmitters.

**Decision letter after peer review:**

Thank you for submitting your article "Rapid purification and metabolomic profiling of SVs from mammalian brain" for consideration by *eLife*. Your article has been reviewed by three peer reviewers, one of whom is a member of our Board of Reviewing Editors, and the evaluation has been overseen by Gary Westbrook as the Senior Editor. The following individual involved in review of your submission has agreed to reveal their identity: Reinhard Jahn (Reviewer #3). The reviewers have discussed the reviews with one another and the Reviewing Editor has drafted this decision to help you prepare a revised submission.

We would like to draw your attention to changes in our revision policy that we have made in response to COVID-19 (https://elifesciences.org/articles/57162). Specifically, when editors judge that a submitted work as a whole belongs in *eLife* but that some conclusions require a modest amount of additional new data, as they do with your paper, we are asking that the manuscript be revised to either limit claims to those supported by data in hand, or to explicitly state that the relevant conclusions require additional supporting data. Our expectation is that the authors will eventually carry out the additional experiments and report on how they affect the relevant conclusions either in a preprint on bioRxiv or medRxiv, or if appropriate, as a Research Advance in *eLife*, either of which would be linked to the original paper.

Summary:

The manuscript from Chantranupong et al. reports a new method for the immune-isolation of SVs based on the expression of a synaptophysin variant containing a HA-tag. This method allows for rapid isolation of SVs with high purity from both neuronal culture and mouse brain. The authors then carried out metabolomics profiling using LC/MS, allowing for detection, among many other metabolites, of neurotransmitters. This study addresses the classical question of SV content that has only become more important in recent years, due in particular to the accumulating evidence for corelease of classical neurotransmitters. The authors demonstrate dependence of the SV contents on the H^+^ pump, and sensitivity to inactivation of the vesicular glutamate or monoamine transporters. The Materials and methods also contain explicit information that will enable others to perform these experiments.

Essential revisions:

1) The authors use complementary mass spectrometry strategies to identify the expected glutamate and GABA in, respectively, excitatory and inhibitory neuron SVs, with no evidence for additional candidate transmitter. This is interesting although not necessarily definitive. Please comment.

2) With regard to specificity, the authors demonstrate that immunoisolation depends on expression of epitope-tagged synaptophysin, and excludes other organelles such as ER and mitochondria. However, many of these membranes are very different from SVs and it would be particularly helpful to know whether the method excludes endosomes, either axonal (such as GLUT4+) or dendritic (transferrin receptor+), which are more closely related to SVs.

3) Related to the previous point, it is difficult to assess the purity of the final SV fraction. Judged by the immunoblots, the only criterium is the loss of some proteins specific for other membranes. Moreover, the EM image shows the presence of larger membrane vesicles, but none of this is quantified. Some suggestions: Quantitative histograms of vesicle sizes in the EM pictures in order to assess the degree of contamination by large vesicles, adding more markers to the WB analysis, in particular for plasma membrane (e.g. Na, Ca-channels) and myelin), immunolabeling of the isolated vesicles for vesicle markers in order to confirm that at least the majority of the vesicles are indeed SVs.

4) One question not addressed specifically concerns yield. The authors do not indicate the yield from immunoisolation, and this will in the end limit their ability to identify novel SV contents, which presumably requires large amounts of material. Thus, it is important to know how much of the tagged protein is not immunoisolated using this more rapid approach. Along the same line, it is surprising that many of the westerns do not show enrichment of SV proteins. Since the HA antibody is apparently cross-linked to the beads, it should be possible to elute the material and at least indicate how much is loaded on the gel relative to the original homogenate, which would provide evidence for enrichment – the legend and Materials and methods do not appear to contain any information about the amount of material loaded (proportion vs. protein).

5) The value of this study depends on the evidence of improvement over previous methods. Since these were less well documented, it is difficult to know how much of an improvement is achieved by the introduction of an exogenous epitope tag over the use of an endogenous SV protein, but previous work was still able to detect accumulation of glutamate. In light of this question, it is important to document the yield as well as the specificity of the new approach.

6) Based on the microscopic images (Figure 1D and E and Figure 1—figure supplement 1), synapsin seems differently distributed (especially in the cell body) after expression of the HA-tagged synaptophysin. Did the authors check if the expression levels and/or distribution of other SV proteins were affected?

7) As with all tagging strategies, the localization of their tagged protein may not be not identical to the endogenous protein. In particular, this applies to the situation where localization in subcellular compartments is beyond standard light microscopy resolution. This could pose a limitation if one is trying to dissect whether there are two types of vesicles that release different neurotransmitters or if two types of neurotransmitter molecules are packaged into one SV, for example. Please comment.

8) Neuronal culture isolations may introduce contamination since it is difficult not also lyse the soma, potentially resulting in Golgi contamination. Please comment.

9) The high-salt washes may disrupt membranes. Were such washes applied prior to cryo-TEM imaging (e.g., Figure 1G)? The small densities may be fragments of SVs. Of course, for metabolomics studies, this would not be so important.

---

## [Author Response]

Essential revisions:1) The authors use complementary mass spectrometry strategies to identify the expected glutamate and GABA in, respectively, excitatory and inhibitory neuron SVs, with no evidence for additional candidate transmitter. This is interesting although not necessarily definitive. Please comment.

Although we did not observe evidence for additional neurotransmitter candidates in the neurons we profiled, it remains an open question whether they are present. In this work, we only investigated the presence of polar metabolites, and we did not assess whether other metabolite classes such as ions and non-polar compounds are specifically enriched within SVs. For instance, zinc has been proposed to be a neurotransmitter at excitatory synapses but was not detectable by the mass spectrometry methods used in this study. Furthermore, if these candidate neurotransmitters are of low abundance, they may not be detected as our workflow has limitations which stem from both the sensitivity of mass spectrometry combined with the low yields of SVs captured. Finally, in this work we only surveyed a limited subset of neurons in culture and in vivo, and more extensive profiling of other brain regions may reveal the presence of putative neurotransmitters.

The results presented here provide a groundwork for further studies to adapt the method for compatibility with other MS detection methods and to further improve yields, both of which may lead to the identification of novel neurotransmitters. We have included a statement on this subject in the second paragraph of the Discussion.

2) With regard to specificity, the authors demonstrate that immunoisolation depends on expression of epitope-tagged synaptophysin, and excludes other organelles such as ER and mitochondria. However, many of these membranes are very different from SVs and it would be particularly helpful to know whether the method excludes endosomes, either axonal (such as GLUT4+) or dendritic (transferrin receptor+), which are more closely related to SVs.

We repeated the SV immunoisolation and profiled our preparations with an expanded set of markers to more completely assess the specificity of our isolations (Figure 1F). We did observe an enrichment of transferrin receptor positive membranes but not GLUT4 positive membranes in our preparations, suggesting that there are dendritic endosomes present. However, the degree of enrichment of these dendritic membranes is much less than for SV membrane markers, as assessed by synaptophysin, synaptotagmin, synaptobrevin, VGLUT1, and SV2A. We have incorporated this finding within the revised text. Improvement in the localization of SV-tag, which, as presented in our manuscript, is mistargeted to the soma and dendrites, may help to alleviate this issue.

3) Related to the previous point, it is difficult to assess the purity of the final SV fraction. Judged by the immunoblots, the only criterium is the loss of some proteins specific for other membranes. Moreover, the EM image shows the presence of larger membrane vesicles, but none of this is quantified. Some suggestions: Quantitative histograms of vesicle sizes in the EM pictures in order to assess the degree of contamination by large vesicles, adding more markers to the WB analysis, in particular for plasma membrane (e.g. Na, Ca-channels) and myelin), immunolabeling of the isolated vesicles for vesicle markers in order to confirm that at least the majority of the vesicles are indeed SVs.

First, we have analyzed EM images of our SV preps and quantified the distribution of the particle sizes. The analysis is shown in Figure 1—figure supplement 1F. We find that a large fraction of the particles (62%) are in the 40-60 nm range, which is within range of previously reported values for SVs. Approximately 13% of the particles are those larger than 70 nm in diameter, which may be consistent with large dense-core vesicles or unknown contaminating organelles. In addition, there are particles smaller than 40 nm which may represent membrane fragments resulting from high-salt washes performed at the end of the immunoisolation to increase the purity of our SV preps, as noted by reviewer comment #9. We have modified our original discussion of the EM results in the Results section to incorporate these considerations.

Although immunolabeling and EM imaging would be ideal to confirm the identity of these particles SVs, we are unable to perform these experiments due to the method we use to free the SVs from the HA beads. Because of the use of an HA-tag containing 9 tandem repeats, the affinity of the tag for the HA antibody is too strong to be competed with HA peptide. To free captured SVs from the beads while preserving their structural integrity for EM imaging, we used a cysteine protease (ficin) which cleaves the immunoglobulin chain of the HA antibody under gentle conditions (37°C incubation in a Tris-based buffer) to release SVs. However, ficin also non-specifically cleaves proteins found on the surface of SVs, as determined by immunoblotting, making us unable to immunolabel the particles for EM imaging. This also points to another caveat of this process – cleaving these SV proteins may alter SV morphology and contribute to the variation in particle sizes observed in the EM images.

To address the specificity of our SV preps, we performed immunoblotting for an expanded set of organellar markers (Figure 1F). We probed for transferrin receptor and GLUT4 to determine if there were dendritic and axonal endosomal membranes present, respectively. We did find a minor degree of dendritic endosome contamination, which we have now noted in the Results. In addition, we profiled for myelin basic protein, Na^+^ K^+^ ATPase, and CaV2.1 calcium channel but did not find any enrichment of these glial and plasma membranes.

4) One question not addressed specifically concerns yield. The authors do not indicate the yield from immunoisolation, and this will in the end limit their ability to identify novel SV contents, which presumably requires large amounts of material. Thus, it is important to know how much of the tagged protein is not immunoisolated using this more rapid approach. Along the same line, it is surprising that many of the westerns do not show enrichment of SV proteins. Since the HA antibody is apparently cross-linked to the beads, it should be possible to elute the material and at least indicate how much is loaded on the gel relative to the original homogenate, which would provide evidence for enrichment--the legend and Materials and methods do not appear to contain any information about the amount of material loaded (proportion vs. protein).

We have included a more extensive analysis of the yield and enrichment of our preparations. First, we have indicated the proportion of input (0.4%) and immunoprecipitate (5%) loaded in the SDS-PAGE gel (Figure 1F legend). These proportions were optimized to result in comparable immunoblot signals in the input and immunoprecipitate, as the input has significantly more protein than the immunoprecipitate.

We also quantified the yield (Figure 1—figure supplement 1G) by calculating how much of the SV material present in the input was captured in the final immunoprecipitate. For these yield calculations, we did not use SV-tag because a large portion of it is mislocalized in the soma and dendrites, and this would lead to an underestimate of yield. Based on three SV proteins – VGLUT1, SV2A, and synaptobrevin – our yields are ~2%, which is consistent with previously developed methods for rapid organellar isolation (Chen et al., 2016). Because our method is focused on speed of isolation to maintain the metabolite content of SVs, we sacrifice yield.

Another metric to enable comparison of our method to previous ones is the degree of enrichment of SV proteins. Thus, we performed quantitative immunoblotting to assess this and included the results in a new panel (Figure 1H). Our enrichment based on synaptobrevin is 10 fold, which is lower than the 20-23 fold value reported in previous SV isolation methods (Ahmed et al., 2013). Further improvements in the method, such as the use of an endogenous tag to more completely tag SV pools, will be implemented in future studies to improve enrichment.

5) The value of this study depends on the evidence of improvement over previous methods. Since these were less well documented, it is difficult to know how much of an improvement is achieved by the introduction of an exogenous epitope tag over the use of an endogenous SV protein, but previous work was still able to detect accumulation of glutamate. In light of this question, it is important to document the yield as well as the specificity of the new approach.

The key advantage of using an exogenous epitope tag vs. an endogenous SV protein is that it can be expressed in genetically defined subpopulations. This enables SV capture from specific cell types from heterogeneous brain tissue and the accurate interrogation of the neurotransmitter profiles of these neurons. We demonstrate this when we express SV-tag in vivo and can determine the neurotransmitter released by cortical, striatal, and dopaminergic neurons. We highlight these advantages in the fourth paragraph of the Introduction and in the first paragraph of the Results.

To address the issues of specificity, we performed an expanded immunoblot analysis and probed for the presence of axonal and dendritic endosomes, glial membranes, and plasma membranes, in addition to previously profiled cellular compartments (Golgi, ER, cytosol, post-synaptic membrane, lysosome). With the exception of a small degree of contamination with dendritic endosomal membrane as indicated by transferrin receptor, our preps are highly specific for SV proteins (Figure 1F). We also documented yield (Figure 1—figure supplement 1G) by calculating the percent of SV protein input that was present in the final isolate. It is low, in the 2% range, which is expected given that our method is optimized for much more rapid capture compared to previous ones. This tradeoff is necessary as our main goal is to preserve the integrity of the metabolite contents of SVs.

6) Based on the microscopic images (Figure 1D and E and Figure 1—figure supplement 1), synapsin seems differently distributed (especially in the cell body) after expression of the HA-tagged synaptophysin. Did the authors check if the expression levels and/or distribution of other SV proteins were affected?

We have included a more representative image of synapsin in uninfected cortical neurons (Figure 1D and E) which indicates that the expression levels and distribution are comparable in infected and uninfected neurons. Furthermore, we have included an additional stain for bassoon, another presynaptic bouton marker (Figure 1—figure supplement 1A) to demonstrate that its localization is also comparable in both uninfected and SV-tag infected neurons.

7) As with all tagging strategies, the localization of their tagged protein may not be not identical to the endogenous protein. In particular, this applies to the situation where localization in subcellular compartments is beyond standard light microscopy resolution. This could pose a limitation if one is trying to dissect whether there are two types of vesicles that release different neurotransmitters or if two types of neurotransmitter molecules are packaged into one SV, for example. Please comment.

Due to the use of an ectopic overexpressed tag, our method is unable to distinguish between these possibilities of corelease. We can only capture the entire SV pool within neurons rather than subclasses. Our method will enable an investigator to establish whether neurons of interest coexpress multiple neurotransmitters. Follow up studies with higher resolution techniques will be necessary to address the nature of this corelease, such as array tomography or immunolabelling with EM imaging. We have included this statement in the fourth paragraph of the Discussion.

8) Neuronal culture isolations may introduce contamination since it is difficult not also lyse the soma, potentially resulting in Golgi contamination. Please comment.

We stained for Golgin-97, a protein marker for golgi membranes, and did not observe enrichment of this protein in our final SV preps (Figure 1F). In our workflow, we generate synaptosomes, which may have helped to minimize somatic contamination.

9) The high-salt washes may disrupt membranes. Were such washes applied prior to cryo-TEM imaging (e.g., Figure 1G)? The small densities may be fragments of SVs. Of course, for metabolomics studies, this would not be so important.

These washes were applied prior to the EM and although we found them to be necessary to minimize contamination (particularly of post-synaptic membranes), this may have led to the disruption of SV membranes. We have mentioned this caveat in the fourth paragraph of the Results. In future studies, we will pursue alternative methods to increase the specificity of our capture and minimize the need for these washes. This includes improving the localization of SV-tag, as mistargeting leads to increased non-specific capture and the requirement for harsh washes.